# Between-species variation in neocortical sulcal anatomy of the carnivoran brain

Magdalena Boch[1,2]*, Katrin Karadachka[3], Kep-Kee Loh[4,5], R Austin Benn[1,6], Lea Roumazeilles[1], Mads F Bertelsen[7], Paul R Manger[8], Ethan Wrigglesworth[9], Simon Spiro[9], Muhammad A Spocter[8,10], Philippa J Johnson[11], Kamilla Avelino-de-Souza[12], Nina Patzke[13,14], Claus Lamm[2], Karla L Miller[1], Jérôme Sallet[1,15], Alexandre A Khrapitchev[16], Benjamin C Tendler[1], Rogier B Mars[1,3]

[1]Oxford University Centre for Integrative Neuroimaging (OxCIN), FMRIB Centre, Nuffield Department of Clinical Neurosciences, University of Oxford, Oxford, United Kingdom; [2]Social, Cognitive and Affective Neuroscience Unit, Department of Cognition, Emotion, and Methods in Psychology, Faculty of Psychology, University of Vienna, Vienna, Austria; [3]Donders Institute for Brain, Cognition and Behaviour, Radboud University Nijmegen, Nijmegen, Netherlands; [4]Montreal Neurological Institute, McGill University, Montreal, Canada; [5]Department of Psychology, National University of Singapore, Singapore, Singapore; [6]Université Paris Cité, CNRS, Integrative Neuroscience and Cognition Center, Paris, France; [7]Copenhagen Zoo, Frederiksberg, Denmark; [8]School of Anatomical Sciences, Faculty of Health Sciences, University of the Witwatersrand, Johannesburg, South Africa; [9]Zoological Society of London, London, United Kingdom; [10]Department of Anatomy, Des Moines University, West Des Moines, United States; [11]Department of Clinical Sciences, Cornell College of Veterinary Medicine, Cornell University, Ithaca, United States; [12]Brazilian Neurobiodiversity Network, Physics Institute, Federal University of Rio de Janeiro, Rio de Janeiro, Brazil; [13]Department of Biological Science, Faculty of Sciences, Hokkaido University, Sapporo, Japan; [14]Faculty of Medicine, IMBB, HMU Health and Medical University, Potsdam, Germany; [15]Univ Lyon, Université Lyon 1, Inserm, Stem Cell and Brain Research Institute, Bron, France; [16]Department of Oncology, University of Oxford, Oxford, United Kingdom

*For correspondence:
magdalena.boch@ndcn.ox.ac.uk

Competing interest: The authors declare that no competing interests exist.

## eLife Assessment

This **valuable** study presents the first detailed and comprehensive description of brain sulcus anatomy of a range of carnivoran species based on a robust manual labeling model allowing species comparisons. The database and method for reconstructing cortical surfaces are **compelling**, and the evidence supporting the conclusions is **solid**. Despite the additional specimen, the evaluation of intra-species variations remains limited, but an insight into the inter-individual variability is now available for certain species. Exploring the associations between sulcal length and behavioral characteristics further suggests the potential of sulci as a proxy of functional organization. Setting an instructive foundation for comparative anatomy, this study will be of interest to neuroscientists and neuroimaging researchers interested in that field, as well as in brain morphology and sulcal patterns, their phylogeny and ontogeny in relation to functional development and behaviour.

**Abstract** Carnivorans are an important study object for comparative neuroscience, as they exhibit a wide range of behaviours, ecological adaptations, and social structures. Previous studies have mainly examined relative brain size, but a comprehensive understanding of brain diversity requires the investigation of other aspects of their neuroanatomy. Here, we obtained primarily post-mortem brain scans from 26 species of the order Carnivora, spanning across eight families with diverse representatives and including additional individuals for selected species, to create the largest carnivoran brain collection to date. We reconstructed their cortical surfaces and examined neocortical sulcal anatomy to establish a framework for systematic interspecies comparisons, revealing distinct regional variations in sulcal anatomy, potentially related to the species' behaviour and ecology. Arctoidea species with pronounced forepaw dexterity exhibited complex sulcal configurations in the presumed somatosensory cortex but low sulcal complexity in the presumed visual and auditory occipitotemporal cortex. Canidae had the largest number of unique major sulci, including one in the occipital cortex and highly social canids featuring an additional frontal cortex sulcus. We also observed differentially complex occipitotemporal sulcal patterns in Felidae and Canidae, indicative of changes in auditory and visual areas that may be related to foraging strategies and social behaviour. In conclusion, this study presents an inventory of the sulcal anatomy of a number of rarely studied carnivoran brains including detailed digital atlases and establishes a framework and novel avenues for further investigations employing a variety of neuroimaging modalities to reveal more about carnivoran brain diversity.

## Introduction

Species of the order Carnivora are a diverse group of mammals comprising at least 270 different species. Carnivorans have colonised various ecological niches, ranging from cursorial and arboreal terrestrial to semi-aquatic and aquatic. They show a range of approaches when interacting with their environments, for example, preferentially using their paws or snouts. Despite their common association with meat eating, carnivorans have a variety of diets, including the completely herbivorous giant panda (*Wilson and Mittermeier, 2009*). They display a wide range of social structures, from solitary to complex hierarchies, and the order contains wild, fully domesticated, and partially feral animals (*Macdonald, 1983*). This diversity in behaviour and ecology, coupled with their relatively large brains (*Smaers et al., 2021*), makes carnivorans a key study object for comparative and evolutionary neuroscience.

Large-scale comparisons of different species' brains have often focused on encephalisation, investigating whether larger relative brain size is associated with particular ecological or social factors (*Healy, 2021*; *Jerison, 1985*). Within carnivorans, this work has suggested that increases in relative brain sizes are predominantly associated with environmental factors, especially in the context of a cognitive buffer against environmental instability during foraging (*Holekamp and Benson-Amram, 2017*; *Michaud et al., 2022*). However, a full understanding of brain diversity and its potential causes and effects requires a focus not only on brain size but also on underlying aspects of brain organisation, including regional variation in size, connections, neurotransmitters, and gene expression (*Krubitzer and Kaas, 2005*; *Mars et al., 2021*). Within primates, such comparisons have provided a substantially more detailed understanding of how brains can differ within a specific lineage (*Hecht et al., 2013*; *Hori et al., 2020*; *Roumazeilles et al., 2020*). Similarly, within carnivorans, different dog breeds show that regional differences in brain size correlate with their behavioural specialisations (*Hecht et al., 2019*), and comparisons between domesticated and wild canids reveal regional differences in gyrification (*Grewal et al., 2020*). Furthermore, novel investigations indicate unique gene expressions in Canidae compared to other mammals (*Sacco et al., 2023*) and suggest a link between neural complexity and cortical folding in Felidae (*Nelson et al., 2024*). This shows that a comprehensive understanding of brain diversity in carnivorans requires us to study multiple aspects of brain organisation.

Here, we present the first in a series of studies exploring the diversity in carnivoran brain organisation, focusing on the identification of the major sulci in surface reconstructions of the neocortex. We exploit the fact that neuroimaging techniques, such as magnetic resonance imaging (MRI), now enable us to collect whole-brain data of various tissue properties quickly, affordably, without damage to the specimen, and in a format that is easily shared between researchers (*Mars et al., 2014*; *Thiebaut de Schotten et al., 2019*). Neocortical sulci present clear anatomical landmarks that can be

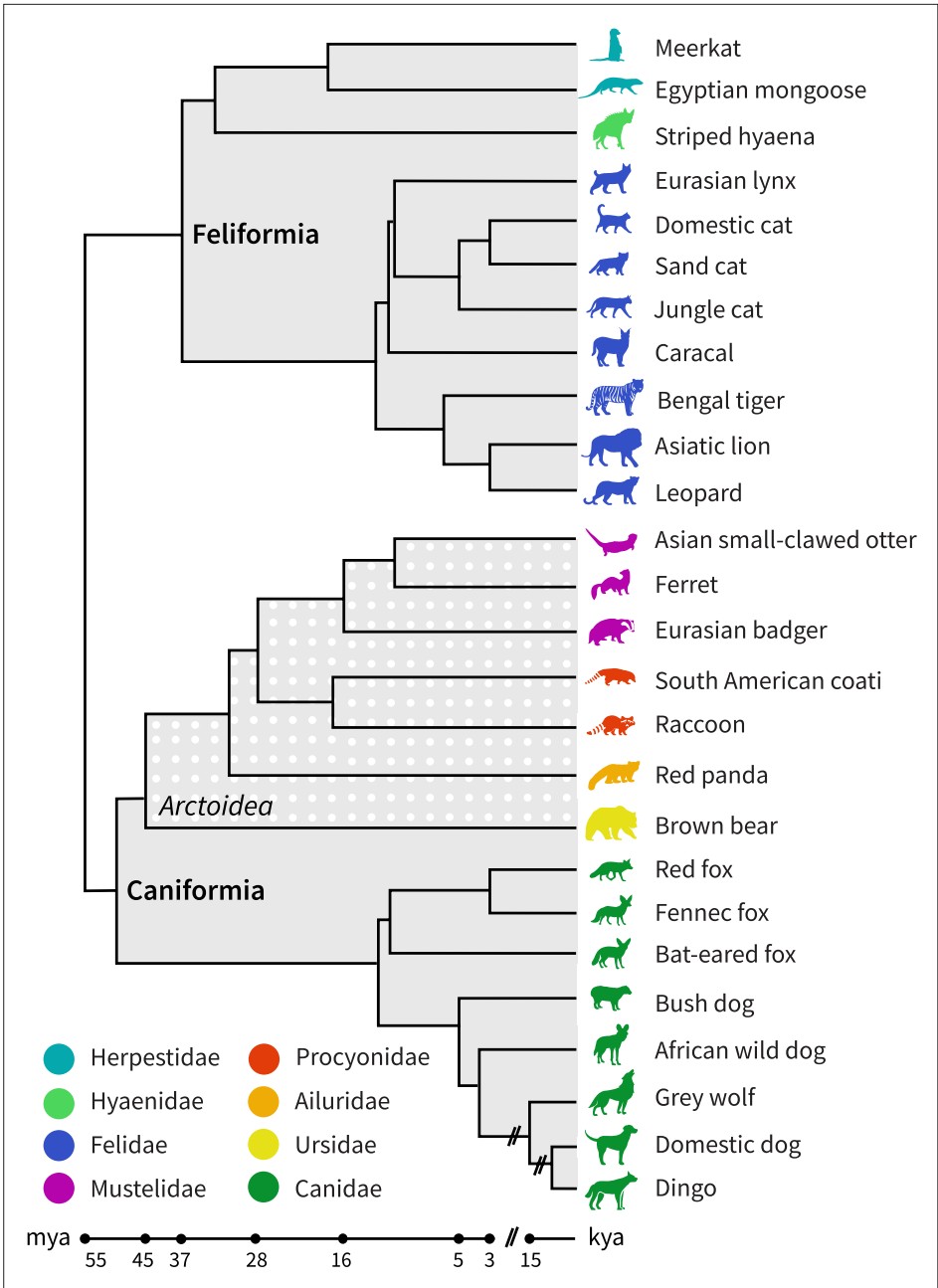

**Figure 1.** Phylogeny of carnivoran sample. The sample consists of species from both carnivoran sub-orders, Caniformia and Feliformia and includes members of eight different families (see also *Table 1* in Materials and methods for a detailed sample description). The sub-order Caniformia includes Canidae and five Arctoidea families. All animals are terrestrial, except for the semi-aquatic Asian small-clawed otter. Phylogenetic relationships were derived from *Field et al., 2022*; *Freedman et al., 2014*; *Kumar et al., 2017*. k/mya, thousand/million years ago.

unambiguously established, providing a frame of reference for navigating the neocortex of individual species, thus allowing macro-level comparison across species, as has been done for primates (*Amiez et al., 2023*; *Connolly, 1950*). Although there are reports of sulcal anatomy in carnivorans, these either focus on individual species (*Chengetanai et al., 2020c*), a small range of species (*Radinsky, 1975b*), or a particular neural system (*Lyras and Van Der Geer, 2003*; *Welker and Campos, 1963*) studied in a few species.

In the present study, we provide a comprehensive overview of neocortical sulcal neuroanatomy across 26 carnivoran species (*Figure 1*). We discuss the key differences in sulcal configurations across the brain, which cortical regions likely expanded in particular lineages, and how this might relate to each species' behaviour and ecology.

## Results and discussion

We labelled the neocortical sulci of 26 carnivoran species (see *Figure 1*) based on reconstructed surfaces and developed standardised criteria (recipes) for identifying each major sulcus. For each sulcus, we also created corresponding digital masks. Our study included 11 Feliformia and 15 Caniformia species from eight different carnivoran families. Within the sub-order Caniformia, we examined eight Canidae and seven Arctoidea species. In addition, we describe relative intra-species variation in sulcal shape based on supplementary specimens from six species (see *Table 1*).

Overall, of the carnivorans studied, Canidae brains exhibited the largest number of unique major sulci, while the brown bear brain was the most gyrencephalic, with the deepest folds and many secondary sulci (see *Figures 2 and 3*; brains are arranged by descending number of major sulci). The brown bear was also the largest animal in the sample. The brains of the smaller species, such as the fennec fox, meerkat or ferret, were the most lissencephalic, with the sulci having fewer undulations or indentations compared to the other species. A similar trend has also been observed in the sulci of the prefrontal cortex in primates (*Amiez et al., 2023*; *Amiez et al., 2019*). The meerkat and Egyptian mongoose exhibited the smallest number of major sulci but possessed, along with the striped hyaena, a unique configuration of sulci in the occipitotemporal cortex. In the following, we describe each sulcus' appearance, the recipes on how to identify them, and provide an overview of the most significant differences across species.

### Identification of major sulci in the carnivoran neocortex

#### Occipitotemporal region

##### Pseudo-sylvian (or sylvian) fissure

The pseudo-sylvian sulcus presents as a fissure on the lateral surface, originating at the ventral border of the neocortex. It extends dorsocaudally, displaying a distinct caudal inclination, and was present in all brains studied (*Figure 2*, cyan). Analogous to the primate sylvian fissure, the insular cortex can be found within the fissure. However, unlike its primate namesake, the carnivoran pseudo-sylvian fissure is centrally located within the temporal lobe, with occipito- and parietotemporal sulci forming concentric or u-shaped arches around it. As the primate and carnivoran temporal lobe are thought to have evolved independently, the primate sylvian and carnivoran pseudo-sylvian fissure should not be considered homologous (*Kaas, 2013*; *Lyras, 2009*). To avoid incorrectly attributing homology between structures, we refer to the fissure in carnivorans as the pseudo-sylvian fissure.

In the majority of carnivoran brains, the pseudo-sylvian fissure curved upwards at its caudal end. Exceptions were noted in the red panda, meerkat, Egyptian mongoose, and striped hyaena, where we observed an open u-shaped fissure. The pseudo-sylvian fissure in the red panda was split at the middle section. However, prior observations indicate that it can also have a continuous u-shape (*England, 1973*), which we observed in the second red panda (*Figure 2—figure supplement 2A*). Preliminary evidence suggests that the insular cortex of the red panda (*Buchanan and Johnson, 2011*) is not fully covered by the opened pseudo-sylvian fissure, which might also be the case for meerkats, Egyptian mongooses and striped hyaenas. We did not observe significant variations in length and shape of the pseudo-sylvian fissure within Canidae and Felidae brains, including when compared to additional specimens of the domestic dog, grey wolf, sand cat, and Persian leopard (*Figure 2—figure supplement 2A*). Compared to Felidae and Canidae, the pseudo-sylvian fissure of the Asian small-clawed otter, Eurasian badger, South American coati, raccoon and brown bear was elongated and extended more dorsally.

##### Ectosylvian sulcus

The ectosylvian sulcus varies significantly in shape and occurrence across Carnivoran species. In Canidae brains, the ectosylvian sulcus is the first concentric sulcus folding around the pseudo-sylvian fissure (*Figure 2*, brown). In Felidae, the sulcus is generally not a continuous arc but is divided into

**Table 1.** Overview carnivoran brain collection.

| Common name | Sub-order | Family | Genus | Species | Sex | Age | Scan protocol | Source |
|---|---|---|---|---|---|---|---|---|
| Red panda | Can | Ailuridae | *Ailurus* | *fulgens* | f | A | Narrow, 7T | 2 |
| Red panda[a] | Can | Ailuridae | *Ailurus* | *fulgens* | f | A | Narrow, 7T | 7 |
| Dingo | Can | Canidae | *Canis* | *dingo* | u | A | Narrow, 7T | 1 |
| Domestic dog[b] | Can | Canidae | *Canis* | *familiaris* | f | A | Narrow, 7T | 6 |
| Domestic dog | Can | Canidae | *Canis* | *familiaris* | n/a | n/a | Wide, 3T | 9 |
| American wolf | Can | Canidae | *Canis* | *lupus* | f | A | Wide, 7T | 1 |
| European wolf[a] | Can | Canidae | *Canis* | *lupus* | m | SA | Wide, 7T | 1 |
| African wild dog | Can | Canidae | *Lycaon* | *pictus* | m | A | Wide, 7T | 1 |
| Bat-eared fox | Can | Canidae | *Otocyon* | *megalotis* | f | A | Wide, 3T | 8 |
| Bush dog | Can | Canidae | *Speothos* | *venaticus* | f | A | Narrow, 7T | 1 |
| Red fox | Can | Canidae | *Vulpes* | *vulpes* | m | S/A | Narrow, 7T | 4 |
| Fennec fox | Can | Canidae | *Vulpes* | *zerda* | m | S/A | Narrow, 7T | 4 |
| Asian small-clawed otter | Can | Mustelidae | *Aonyx* | *cinereus* | m | A | Narrow, 7T | 1 |
| Eurasian badger | Can | Mustelidae | *Meles* | *meles* | f | A | Narrow, 7T | 2 |
| Ferret | Can | Mustelidae | *Mustela* | *putorius* | u | u | Narrow, 7T | 7 |
| South American coati | Can | Procyonidae | *Nasua* | *nasua* | m | SA | Narrow, 7T | 1 |
| Raccoon | Can | Procyonidae | *Procyon* | *lotor* | m | A | Wide, 3T | 5 |
| Brown bear | Can | Ursidae | *Ursus* | *Arctos (arctos)* | f | SA | Wide, 7T | 1 |
| Ussuri brown bear[a] | Can | Ursidae | *Ursus* | *arctos (lasiotus)* | f | SA | Wide, 3T | 5 |
| Caracal | Fel | Felidae | *Caracal* | caracal | u | u | Narrow, 7T | 7 |
| Domestic cat[c] | Fel | Felidae | *Felis* | *catus* | f | SA | Wide, 3T | 3 |
| Jungle cat | Fel | Felidae | *Felis* | *chaus* | u | u | Narrow, 7T | 7 |
| Sand cat | Fel | Felidae | *Felis* | *margarita* | u | u | Narrow, 7T | 7 |
| Arabic sand cat[a] | Fel | Felidae | *Felis* | *margarita (harrisoni)* | f | A | Wide, 3T | 8 |
| Eurasian lynx | Fel | Felidae | *Lynx* | *lynx* | m | A | Narrow, 7T | 2 |
| Asiatic lion | Fel | Felidae | *Panthera* | *leo* | f | SA | Wide, 7T | 1 |
| Amur leopard | Fel | Felidae | *Panthera* | *pardus (orientalis)* | m | SA | Wide, 7T | 1 |
| Persian leopard[a] | Fel | Felidae | *Panthera* | *pardus (tulliana)* | u | u | Wide, 3T | 7 |
| Bengal tiger | Fel | Felidae | *Panthera* | *tigris* | u | u | Wide, 3T | 7 |
| Egyptian mongoose | Fel | Herpestidae | *Herpestes* | *ichneumon* | u | u | Narrow, 7T | 7 |
| Meerkat | Fel | Herpestidae | *Suricata* | *suricatta* | m | A | Narrow, 7T | 2 |
| Striped hyaena | Fel | Hyenidae | *Hyaena* | *hyaena* | u | u | Narrow, 7T | 7 |

Note. All animals are adults, *n* = 14 species belong to the Carnivoran sub-order Feliformia (Fel), and *n* = 16 are Caniformia (Can) species. Overall, our sample comprises eight different Carnivoran families. All data, except for the domestic cat, are post-mortem samples, and data were obtained from multiple sources: 1 = Copenhagen Zoo specimen collection, 2 = the Zoological Society of London, 3 = the Cornell University College of Veterinary Medicine, 4 = St. Louis Zoological Gardens, 5 = Hokkaido University (Hokkaido, Japan), 6 = donated by a private owner, 7 = Mammalian MRI (MaMI) database (**Assaf et al., 2020**), 8 = Lyon Comparative Brain Collection acquired by J.S. and colleagues, 9 = stereotactic breed-averaged template (**Johnson et al., 2020**). Data was collected with wide-bore scanners, typically used for human neuroimaging, or narrow-bore scanners, typically used for rodent neuroimaging. Field strength of the scanners was either 3 or 7 tesla (T; see scan protocol). [a]For consistency, we only labelled one specimen in the main study, marked (sub-)species are added for supplementary analyses. [b]The domestic dog sample is a Belgian shepherd; we also created a surface from a stereotactic breed-averaged template (**Johnson et al., 2020**) for supplementary confirmatory analysis. [c]The domestic cat sample is a domestic shorthair cat. f, female; m, male; u, unknown; A, adult; SA, subadult; S/A, adult or subadult.

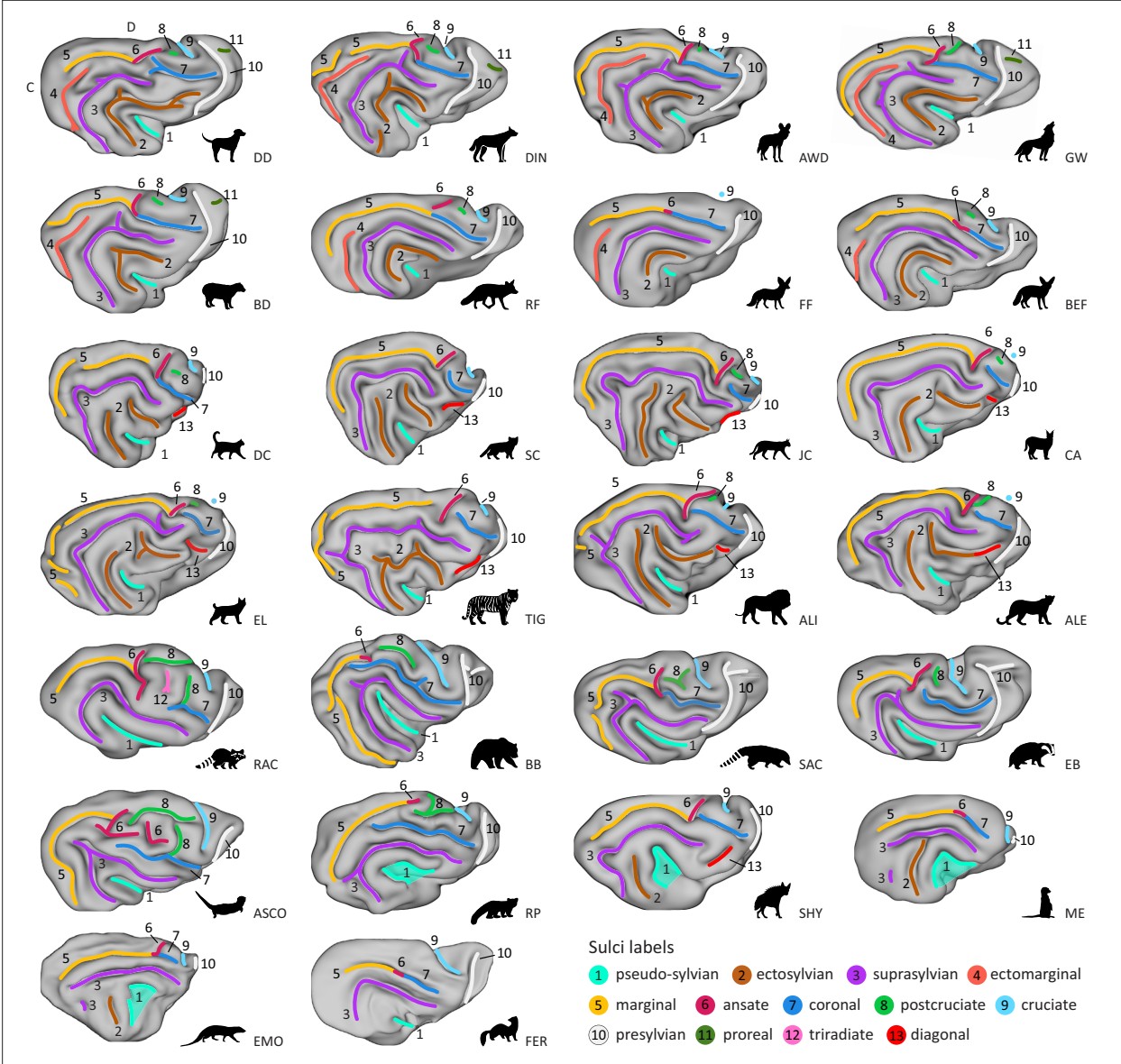

**Figure 2.** Lateral view showing variation of sulcal patterns across carnivorans. Surfaces are presented in descending order of sulcal complexity, starting with the canid brains that exhibited the highest number of unique major sulci. They are followed by felid brains which lacked the ectomarginal (orange, 4) and proreal (dark green, 11) sulcus but exhibited an additional diagonal (red, 13), and a split ectosylvian (brown, 2) sulcus, progressing towards species with the least complex sulcal topology. Despite exhibiting low sulcal complexity, the meerkat, Egyptian mongoose and striped hyaena had a caudal ectosylvian sulcus. The striped hyaena also exhibited a diagonal sulcus, analogous to Felids. All Arctoidea species in the fifth row, and the Asian small-clawed otter and red panda (sixth row, left) had complex or extended cruciate (light blue, 9), postcruciate (light green, 8) and ansate (purple red, 6) sulci. They also exhibited an inverted u-shaped suprasylvian sulcus compared to the arc shape observed in Canids and Felids (top four rows). We indicate the location of the cruciate sulcus with a blue dot if it is not well visible on the lateral surface due to its shape; see *Figure 3* for a dorsal view. Anatomical locations are indicated on the dog surface in the top row, left corner. C, caudal; D, dorsal. ALI, Asiatic lion; ALE, Amur leopard; ASCO, Asian small-clawed otter; AWD, African wild dog; BB, brown bear; BD, bush dog; BEF, bat-eared fox; CA, caracal; DC, domestic cat; DD, domestic dog; DIN, dingo; EB, Eurasian badger; EL, Eurasian lynx; EMO, Egyptian mongoose; FER, ferret; FF, fennec fox; GW, grey wolf; JC, jungle cat; ME, meerkat; RAC, raccoon; RF, red fox; RP, red panda; SAC, South American coati; SC, sand cat; SHY, striped hyaena; TIG, Bengal tiger.

The online version of this article includes the following figure supplement(s) for figure 2:

**Figure supplement 1.** Ventral view of major neocortical sulci.

**Figure supplement 2.** Lateral and ventral view of major neocortical sulci in additional individuals and sub-species.

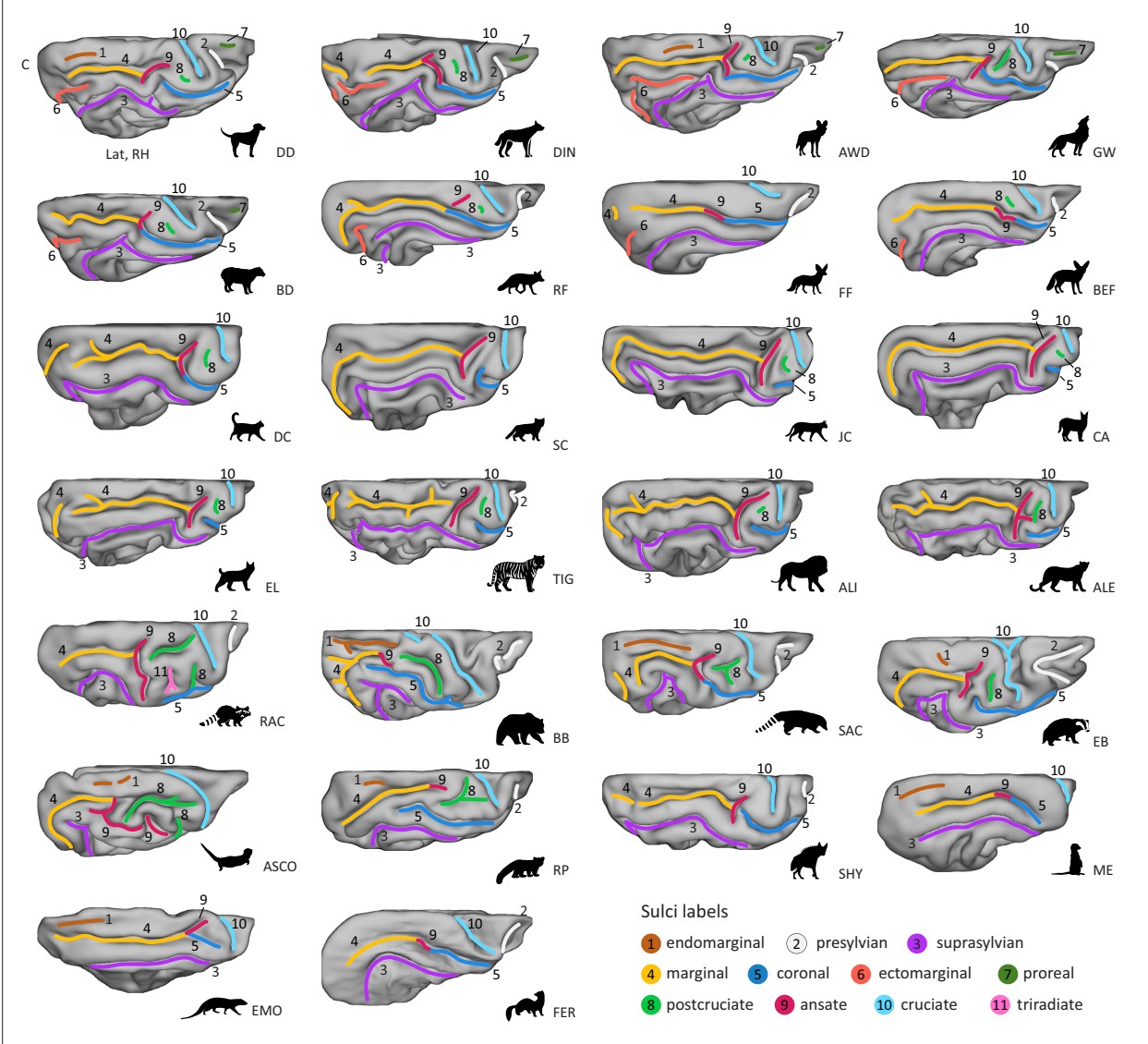

**Figure 3.** Dorsal view reveals varying complexity of sulci in parietal and frontal cortices across carnivoran families. Surfaces are presented in descending order of sulcal complexity, starting with brains exhibiting the most complex sulcal topology and progressing to those with the least, as in *Figure 2*. Only the wolf-like Canids (top row) and the bush dog (second row, left) had a proreal sulcus (green, 7). The brown bear had a secondary cruciate sulcus (light blue, 10). Anatomical locations are indicated on the domestic dog surface in the top row, left corner. Lat, lateral; C, caudal; RH, right hemisphere. ALI, Asiatic lion; ALE, Amur leopard; ASCO, Asian small-clawed otter; AWD, African wild dog; BB, brown bear; BD, bush dog; BEF, bat-eared fox; CA, caracal; DC, domestic cat; DD, domestic dog; DIN, dingo; EB, Eurasian badger; EL, Eurasian lynx; EMO, Egyptian mongoose; FER, ferret; FF, fennec fox; GW, grey wolf; JC, jungle cat; ME, meerkat; RAC, raccoon; RF, red fox; RP, red panda; SAC, South American coati; SC, sand cat; SHY, striped hyaena; TIG, Bengal tiger.

The online version of this article includes the following figure supplement(s) for figure 3:

**Figure supplement 1.** Major neocortical sulci in the medial wall.

**Figure supplement 2.** Dorsal and medial view of major neocortical sulci in additional individuals and sub-species.

a caudal and rostral ectosylvian sulcus; only in lions has a study reported variable occurrences of a continuous arc (*Sakai et al., 2016*).

In our sample, we observed a divided ectosylvian sulcus in all Felidae, except for the Bengal tiger, which exhibited a continuous sulcus on the right hemisphere with a dent in the middle part, and a divided sulcus on the left. All Canidae displayed a continuous arc, except the dingo, in which the caudal end of the ectosylvian sulcus was split into two parts. The meerkat, Egyptian mongoose and striped hyaena only exhibited a caudal ectosylvian sulcus, and the Asian small-clawed otter, Eurasian

badger, ferret, South American coati, raccoon, red panda, and brown bear (i.e., all Arctoidea species) brains did not have an ectosylvian sulcus. None of the supplementary samples revealed major variations in shape or presence of the sulcus (*Figure 2—figure supplement 2B*).

## Suprasylvian sulcus

The suprasylvian sulcus is present in all carnivorans with only slight variations (*Figures 2 and 3*, purple). In Canidae, it represents the second arc curving around the pseudo-sylvian fissure; in all other species, it is the first complete arc.

In the fennec fox, meerkat, Egyptian mongoose, and ferret, the suprasylvian sulcus was smoother, with no radial branches arising from the sulcus and a smoother overall curvature as was observed in the other species. In the meerkat and Egyptian mongoose, the caudal end of the sulcus was also less developed and did not bend, but we observed a detached indentation where the caudal bend was observed in other species. Together with the caudal ectosylvian sulcus, the suprasylvian sulcus formed the only (incomplete) arc in these two brains. In the brown bear, Asian small-clawed otter, Eurasian badger, and South American coati, the sulcus had an inverted u-shaped form. Observations in the supplementary samples were consistent with these patterns and did not reveal any major variation in the shape or presence of the sulcus (*Figure 2—figure supplement 2A*).

## Ectomarginal (or ectolateral) sulcus

The ectomarginal sulcus is only present in the canid brains and runs between the marginal and suprasylvian sulcus (*Figures 2 and 3*, orange). In all canids, including the supplementary samples (*Figure 2—figure supplement 2A*, *Figure 3—figure supplement 2A*), the sulcus had a curved pattern similar to the suprasylvian and ectosylvian sulci, but it formed only the caudal half of a complete arc.

## Marginal (or lateral, intraparietal) sulcus

The marginal sulcus is an extensive longitudinal sulcus that runs anteriorly from the occipital lobe along the dorsal convexity parallel to the midline and often merges or superficially connects with the perpendicularly directed ansate sulcus that lies caudal to the postcruciate and cruciate sulcus in the parietal lobe (*Figures 2 and 3*, yellow). The occasionally used term 'intraparietal' is suggestive of potential homology with the primate intraparietal sulcus of the same name, but it should be noted that the carnivoran parietal cortex is significantly smaller than that of the primate (*Garin et al., 2022*), and thus, the carnivoran sulcus' territory encompasses other anatomical areas.

The curvature and caudal extent of the marginal sulcus varied across species. It was relatively short and minimally curved in the meerkat, Egyptian mongoose, ferret, and fennec fox. In other species, the caudal end was more elongated, curving roughly parallel to the suprasylvian, or the ectomarginal sulcus in Canidae. In the brown bear, Asian small-clawed otter, and Eurasian badger, the caudal portion of the marginal sulcus extended even more ventrally than in the other species and almost formed an additional incomplete arc with the ansate and coronal sulcus. In the brown bear and Ussuri brown bear, the marginal sulcus extended the most ventrally and terminated close to the caudal rhinal fissure (see Appendix—Supplementary Note 1 and *Figure 2—figure supplements 1 and 2B* for ventral view).

In several species, including the dingo, domestic cat, brown bear, and South American coati and further supplementary individuals (*Figure 2—figure supplement 2B*), the caudal portion of the marginal sulcus was detached in one or both hemispheres, which is a frequently reported occurrence (*England, 1973*; *Kawamura and Naito, 1978*). Potentially due to the similar caudal bend, some authors have labelled the (detached) caudal portion of the marginal sulcus in Ursidae as the ectomarginal sulcus (*Lyras et al., 2023*, but see e.g., *Sienkiewicz et al., 2019*).

The (detached) caudal marginal sulcus in Ursidae continues the course of the marginal sulcus caudally and/or ventrally and is topologically continuous with it. In contrast, the ectomarginal sulcus in Canidae is an entirely separate sulcus that runs between the suprasylvian and marginal sulci, forming a small, additional arch that is rarely connected to the marginal sulcus (*Kawamura and Naito, 1978*). This distinction is illustrated, for example, in the dingo and grey wolf. In the dingo, we observed both a detached caudal extension of the marginal sulcus and a distinct ectomarginal sulcus. In both grey wolf specimens, the marginal sulcus extended ventrally in a way that resembled the brown bear, but they also exhibited a clearly separate ectomarginal sulcus,

confirming that the two features are not equivalent. In contrast, in the brown bear and Ussuri brown bear (***Figure 2—figure supplement 2B***), we observed variation in whether the marginal sulcus was detached or continuous, but no separate sulcus resembling the ectomarginal sulcus seen in Canidae.

## Endomarginal (or parietal, entolateral, endolateral)

The endomarginal sulcus runs between the marginal sulcus and the median longitudinal fissure on the dorsal surface of the carnivoran brain (***Figure 3***, brown).

Although the endomarginal sulcus has been reported to be present in larger-sized wolf-like canids (***Lyras, 2009***; ***Radinsky, 1969***), in the present study, it was only identified in the African wild dog and the domestic dog (but not in the breed-averaged template; ***Figure 3—figure supplement 2A***). Prior observations in domestic dogs confirm that the endomarginal sulcus is not always readily identifiable (***Czeibert et al., 2018***).

The brown bear and South American coati had the longest endomarginal sulcus, which is occasionally also referred to as 'parietal sulcus' in the brown bear (***Sienkiewicz et al., 2019***). In the Asian small-clawed otter, the endomarginal sulcus was split into two parts, but as shown previously, it can also appear as a continuous sulcus (***Radinsky, 1968***). The meerkat and Egyptian mongoose had a continuous endomarginal sulcus, which has also been termed the posterior marginal sulcus (***Radinsky, 1975b***), but due to its position and shape in both brains, we refer to it as the endomarginal sulcus. The endomarginal sulcus of the Eurasian badger presented as a small groove; the raccoon, striped hyaena, ferret, and felids did not have an endomarginal sulcus.

## Fronto-parietal region

### Diagonal sulcus

The diagonal sulcus is oriented nearly perpendicularly to the rostral portion of the suprasylvian sulcus (***Figure 3***, ***Figure 2—figure supplement 1***, red). We identified it in all Felidae and in the striped hyaena, but it was absent in Herpestidae and all Caniformia species.

In our sample, the sulcus showed moderate variation in shape and continuity. In the caracal and the second sand cat, it appeared as a detached continuation of the rostral suprasylvian sulcus (***Figure 2—figure supplement 2***). In the Amur and Persian leopards, the diagonal sulcus merged with the rostral ectosylvian sulcus on the right hemisphere, forming a continuous or bifurcated groove. Similar individual variation has been described in domestic cats (***Kawamura, 1971***).

### Ansate and coronal sulcus

The rostral end of the marginal sulcus coincides with a complexly organised sulcal region comprising two distinct sulci. The first sulcus runs dorsomedially towards the medial longitudinal fissure and is named the ansate sulcus (***Figures 2 and 3***, purple red). The second sulcus continues rostrally on the lateral surface, curving around the cruciate sulcus, and is termed the coronal sulcus (***Figures 2 and 3***, dark blue).

In the majority of canids, as well as in Herpestidae and the striped hyaena (left hemisphere), the coronal sulcus merged with the ansate and marginal sulcus. In Felidae and Arctoidea species, the coronal sulcus was primarily detached. The coronal sulcus of all Arctoidea species extended more laterally; in the brown bear, Asian small-clawed otter, and red panda, it was also distinctly more elongated compared to the other species.

The raccoon exhibited the most elongated ansate sulcus, and in the Asian small-clawed otter, it was the most complex, with multiple perpendicular branches extending in the region between the postcruciate and coronal sulcus. Based on its position and shape, the detached rostral portion of the ansate sulcus in the Asian small-clawed otter resembles the triradiate sulcus of raccoons (***Figure 2***, light pink), but there are also reports of a continuous sulcus (***Welker and Campos, 1963***). Conversely, in the fennec fox, bush dog, red panda, meerkat, ferret, brown bear (but not the Ussuri brown bear), and European wolf (***Figure 2***, ***Figure 2—figure supplement 2A***), the ansate sulcus was more difficult to identify, lacking a clear perpendicular form. Here, we labelled the dorsal bend at the rostral end of the marginal sulcus as the ansate sulcus.

## Cruciate, postcruciate, and triradiate sulcus

The cruciate sulcus (*Figures 2 and 3*, light blue) is positioned roughly orthogonal to the median longitudinal fissure on the dorsal convexity of the brain. On the medial wall, it often merges with the dorsal longitudinal splenial sulcus (see Appendix—Supplementary Note 2 and *Figure 3—figure supplement 1* for medial view). The postcruciate sulcus (*Figures 2 and 3*, light green) is located between the coronal/ansate and the cruciate sulcus. The triradiate sulcus (*Figures 2 and 3*, light pink) is a raccoon speciality (*Welker and Campos, 1963*; *Welker and Seidenstein, 1959*) situated between the ansate, postcruciate, and coronal sulcus.

The cruciate sulcus of the Felidae species examined and the striped hyaena extended orthogonally from the longitudinal fissure and stayed mostly on the dorsal surface of the brain and was, therefore, not clearly visible from a lateral perspective (*Figure 2*). In all other species, the sulcus originated at the midline and exhibited a ventrorostral orientation with bends of varying degrees. The cruciate sulcus of the brown bear, Asian small-clawed otter, Eurasian badger, raccoon, and South American coati was the most elongated and curved ventrally. The brown bear and Bengal tiger also had a short secondary sulcus caudal to the cruciate sulcus, whereas the fennec fox exhibited the least expressed and shortest cruciate sulcus.

Together with the red panda, the species with the most elongated cruciate sulcus (brown bear, Asian small-clawed otter, Eurasian badger, raccoon, and South American coati) also had an extended postcruciate sulcus coupled with a more lateral coronal sulcus (as described above). Due to its complex appearance, the postcruciate sulcus of the red panda and some Procyonidae species, such as the South American coati and raccoon, is often termed the 'postcruciate complex' (*Welker and Campos, 1963*; *Welker and Seidenstein, 1959*). In the second red panda, the ansate sulcus was split into two branches that terminated in close proximity, instead of forming a single branched sulcus (*Figure 2—figure supplement 2A*). The Asian small-clawed otter and raccoon had a dorsal postcruciate sulcus running parallel to the midline with a more ventral portion that merged with the perpendicular coronal sulcus. In these cases, the two parts terminated close to each other, but, as observed previously, they occasionally merge (*Welker and Campos, 1963*; *Welker and Seidenstein, 1959*). In canids and felids, the postcruciate sulcus was primarily a shallow groove and, therefore, not clearly identifiable in all species. For example, in the fennec fox, we noted the postcruciate sulcus only in the left hemisphere, and for the domestic dog only in the right hemisphere, but bilaterally in the dog template (*Figure 3—figure supplement 2*). Of the Felidae and Canidae, the American wolf and amur leopard had the most expressed postcruciate sulcus; in the latter, it merged with a secondary branch of the ansate sulcus. The meerkat, Egyptian mongoose, striped hyaena, ferret, and both sand cats did not have a postcruciate sulcus.

## Presylvian sulcus

Rostral to the pseudo-sylvian fissure, the perisylvian sulcus originates from or close to the rostral lateral rhinal fissure (see Appendix—Supplementary Note 1 and *Figure 2—figure supplement 1* for ventral view). The sulcus extends dorsally, and we observed a gentle caudal curve in the majority of the species (*Figures 2 and 3*, white).

There were no major variations across species, but we noted a shortened sulcus in the meerkat and Egyptian mongoose and the presence of a secondary branch at the dorsal end that extended rostrally in the Eurasian badger and South American coati brain. The brown bear exhibited an additional sulcus in the frontal lobe, previously labelled as the proreal sulcus (see e.g., *Sienkiewicz et al., 2019*); however, its shape closely resembled the secondary branches of the perisylvian sulcus seen in the South American coati and Eurasian badger. *Sienkiewicz et al., 2019* also noted that this sulcus merges with the presylvian sulcus in their specimen, consistent with our findings in the left hemisphere of the brown bear and bilaterally in the Ussuri brown bear (see *Figure 2—figure supplement 2A*, *Figure 3—figure supplement 2A*). Given the known gyrencephaly of Ursidae brains with frequent secondary and tertiary sulci (*Lyras et al., 2023*), we propose that this sulcus represents a branch of the perisylvian sulcus.

## Proreal sulcus

The proreal sulcus is the most rostral sulcus; it is located in the frontal lobe and has an axis that runs orthogonal to the presylvian sulcus (*Figures 2 and 3*, orange).

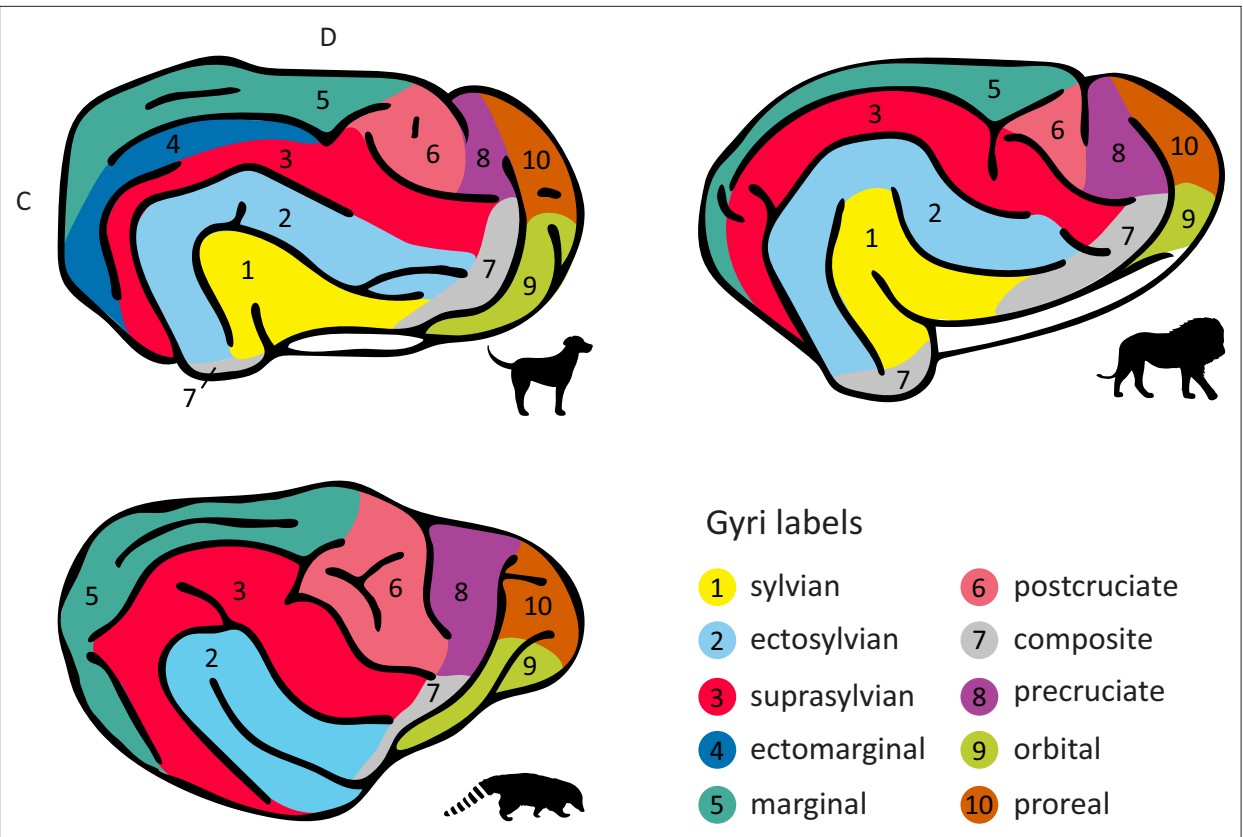

**Figure 4.** Sulcal anatomy variations and corresponding gyral differences. As illustrated on the lateral view of the domestic dog brain (upper left corner), only Canidae have an ectomarginal gyrus (dark blue). Additionally, Canidae and Felidae (Asiatic lion, upper right corner) species have an (incomplete) sylvian gyrus (yellow). Species exhibiting a more complex postcruciate sulcus have an expanded postcruciate gyrus (light red) as illustrated on the South American coati brain (lower left corner). Nomenclature follows prior descriptions in Canidae and Felidae (*Czeibert et al., 2019*; *Pakozdy et al., 2015*; *Rogers Flattery et al., 2023*; *Stolzberg et al., 2017*), and descriptions of the ferret brain for species lacking an ectosylvian sulcus (*Radtke-Schuller, 2018*). The rostral suprasylvian gyrus (red) is also called the coronal gyrus in ferrets, and in the field of paleoneurology, post- and precruciate gyri (light red, purple) are often referred to as the posterior and anterior sigmoid gyrus (see e.g., *Lyras et al., 2023*). White space indicates allocortex. C, caudal; D, dorsal.

This sulcus was present in all canids, including the supplementary samples, except for the red, bat-eared, and fennec fox (i.e., fox-like canids; see also *Figure 3—figure supplement 2A*). The sulcus appeared shallow in the majority of the species.

## Sulcal anatomy variations and corresponding gyral differences

Canidae were the only species where an ectomarginal sulcus could be identified, resulting in the formation of the ectomarginal gyrus situated between the marginal and ectomarginal sulcus (*Figure 4*, dark blue). Additionally, both canids and felids had a (sometimes incomplete) ectosylvian sulcus, defining the outer boundary of the sylvian gyrus, which curves around the pseudo-sylvian fissure (*Figure 4*, yellow). Consequently, species without an ectosylvian sulcus (i.e., Arctoidea) also lacked a sylvian gyrus. Intuitively, one would assume that they lacked the *ecto*sylvian gyrus; however, in the ferret, which lacks an ectosylvian sulcus, the first gyrus curving around the pseudo-sylvian fissure is consistently referred to as the ectosylvian gyrus (see e.g., *Radtke-Schuller, 2018*). This terminology might be attributed to the cortex forming this gyrus housing the primary auditory cortex, a feature shared with the ectosylvian gyrus of the domestic cat and dog or African wild dog (*Bizley et al., 2005*; *Chengetanai et al., 2020a*; *Kosmal, 2000*; *Stolzberg et al., 2017*; and see next section).

In the meerkat and Egyptian mongoose, the caudal ectosylvian sulcus and suprasylvian sulcus form an incomplete arc, which defines the outer boundary of a gyrus curving around the pseudo-sylvian fissure. The suprasylvian sulcus extends further caudal to the incomplete arc and defines the dorsal border of an additional gyrus caudal to the caudal ectosylvian sulcus. In the striped hyaena, the caudal

ectosylvian sulcus forms an additional incomplete arc which divides the gyrus between the caudal ectosylvian and pseudo-sylvian fissure into two parts. To determine if these gyri should be labelled sylvian and/or ectosylvian, investigations into the location of the primary auditory cortex in Herpestidae and Hyenidae are required.

The presence of the proreal sulcus in wolf-like canids and the greater relative complexity and size of the postcruciate sulcus in the red panda, South American coati, raccoon, Eurasian badger, Asian small-clawed otter, and brown bear likely coincide with an expansion of the proreal gyrus in Canidae and of the postcruciate gyrus in Arctoidea (*Figure 4*, brown, light red).

## Lineage-specific observations and potential relationship to function

The understanding of neocortical areas in the carnivoran brain, such as the location of specific sensory regions, remains incomplete or limited in many species. However, by using the sulcal patterns as a framework and combining them with the knowledge we have about sensory regions in some of the species, we can make inferences about the possible expansion of specific sensory regions and form predictions about their location in less frequent or previously unstudied animals. Furthermore, relating the relative complexity of sulcal topology to the animals' behavioural and social ecology provides clues regarding potential drivers of neuroanatomical diversity across species.

### Somatosensory cortex

One significant variation observed across the species studied was the configuration of the postcruciate and cruciate sulci (*Figure 5A*). Compared to all other species, the Asian small-clawed otter, raccoon, South American coati, red panda, and brown bear had an expanded postcruciate sulcus with secondary branches and an elongated cruciate sulcus. The Eurasian badger and ferret had an elongated cruciate sulcus, while the raccoon possessed an additional sulcus, the triradiate sulcus.

Prior electrophysiological recordings and histological investigations in the domestic cat (*Dykes et al., 1980*), domestic dog (*Bromiley et al., 1956*), raccoon, coati, red panda (*Welker and Campos, 1963*; *Welker and Seidenstein, 1959*), and ferret (*Mclaughlin et al., 1998*) have revealed that the posterior region of the neocortex situated between the ansate, cruciate, and suprasylvian sulcus, the postcruciate (or posterior sigmoid) gyrus, is the location of the primary somatosensory cortex (S1; see *Figure 6* for schematic drawings). Where present, the postcruciate sulcus marks the anterior border of S1 within the postcruciate gyrus. S1 further extends ventrally to the rostral suprasylvian sulcus, covering the rostral suprasylvian (or coronal) gyrus, primarily involved in processing tactile information from the head, which also applies to the region surrounding the diagonal sulcus of the domestic cat (*Hess et al., 1952*).

In the raccoon, red panda, coati, and ferret, considerably larger portions of the postcruciate gyrus S1 area appeared to be allocated to representing the forepaw and forelimbs (*Mclaughlin et al., 1998*; *Welker and Campos, 1963*; *Welker and Seidenstein, 1959*) when compared to the domestic cat or dog (*Dykes et al., 1980*; *Bromiley et al., 1956*). This aligns with the observation that all species in the present sample with more complex or elongated postcruciate and cruciate sulci configurations display a preference for using their forepaws when manipulating their environment (see e.g., *Iwaniuk et al., 1999*; *Iwaniuk and Whishaw, 1999*; *Radinsky, 1968 Figure 5A*). Complementary quantitative analyses further support this link, revealing a positive relationship between the relative length of the cruciate and postcruciate sulci and high forepaw dexterity (see *Figure 5—figure supplement 1*, *Figure 5—source data 2 and 3*). This is suggestive of a potential link between sulcal morphology and a behavioural specialisation in Arctoidea, consistent with earlier observations in otter species (*Radinsky, 1968*).

### Occipitotemporal cortical territories

We also observed significant variations in the sulcal topology of the occipitotemporal cortex across species (see *Figure 5B*). In the Caniformia, the Arctoidea species presented with the relatively least complex sulcal topology in the temporal cortex, with only a single sulcal arc, the suprasylvian sulcus (*Figure 6*), being noted. In contrast, all canids exhibited the most complex sulcal configuration featuring a second complete arc, the ectosylvian sulcus, in the temporal cortex and the additional ectomarginal sulcus in the occipital cortex. All felids had an ectosylvian sulcus but lacked the middle part present in Canidae and, therefore, only exhibited an incomplete second arc. Studies of digital

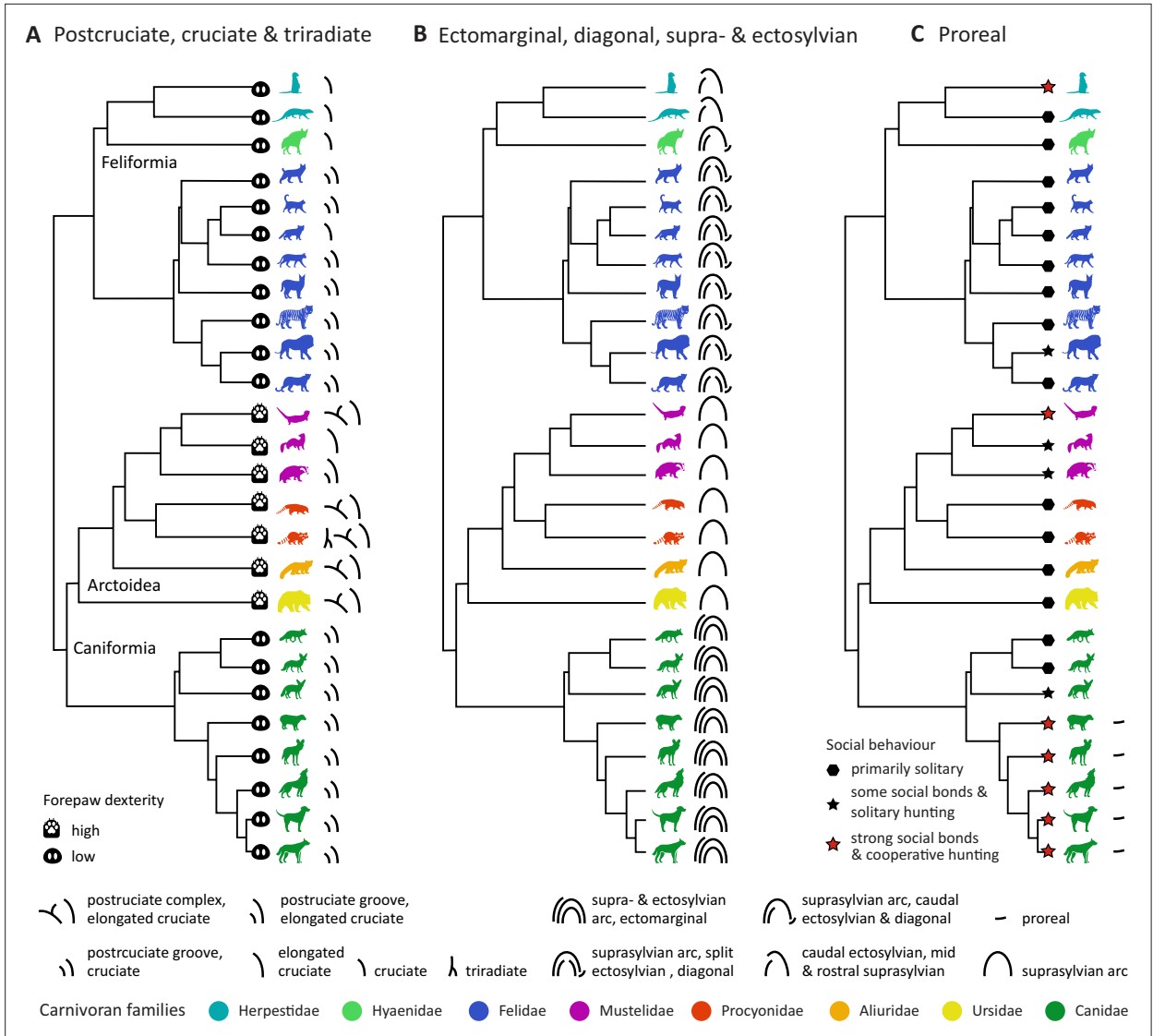

**Figure 5.** Lineage-specific observations and potential functional correlates. (**A**) The Arctoidea species with complex sulcal configurations in the somatosensory cortex also exhibit more pronounced forepaw dexterity (see e.g., *Iwaniuk et al., 1999*; *Iwaniuk and Whishaw, 1999*; *Radinsky, 1968*). In the red panda, coati, and raccoon, this potentially expanded cortical territory surrounding the postcruciate sulcus or complex appears to accommodate the primary somatosensory cortex, with a significant representation of the forepaw (see *Figure 6*). (**B**) Canids had the most complex occipitotemporal sulcal anatomy, followed by felids and the herpestids and striped hyaena with a unique sulcal configuration. This might indicate the expansion of auditory and visual regions in canids and of auditory regions in felids (see *Figure 6*). Arctoidea species exhibited the least complex occipitotemporal sulcal topology. (**C**) All canids with strong social bonds and that engage in cooperative hunting (*Macdonald and Sillero-Zubiri, 2004*; *Wilson and Mittermeier, 2009*; *Wilson, 2000*) had an additional sulcus in the frontal lobe, the proreal sulcus. Complementary quantitative analyses confirmed a positive association between sociality and proreal sulcus length, and between forepaw dexterity and the length of the postcruciate and cruciate sulci (*Figure 5—figure supplement 1*).

The online version of this article includes the following source data and figure supplement(s) for figure 5:

**Source data 1.** Effects of forepaw dexterity and sociality on relative length of the proreal sulcus.

**Source data 2.** Effects of forepaw dexterity and sociality on relative length of the postcruciate sulcus.

**Source data 3.** Effects of forepaw dexterity and sociality on relative length of the cruciate sulcus.

**Figure supplement 1.** Proportion of significant associations between relative sulcal length and behavioural characteristics.

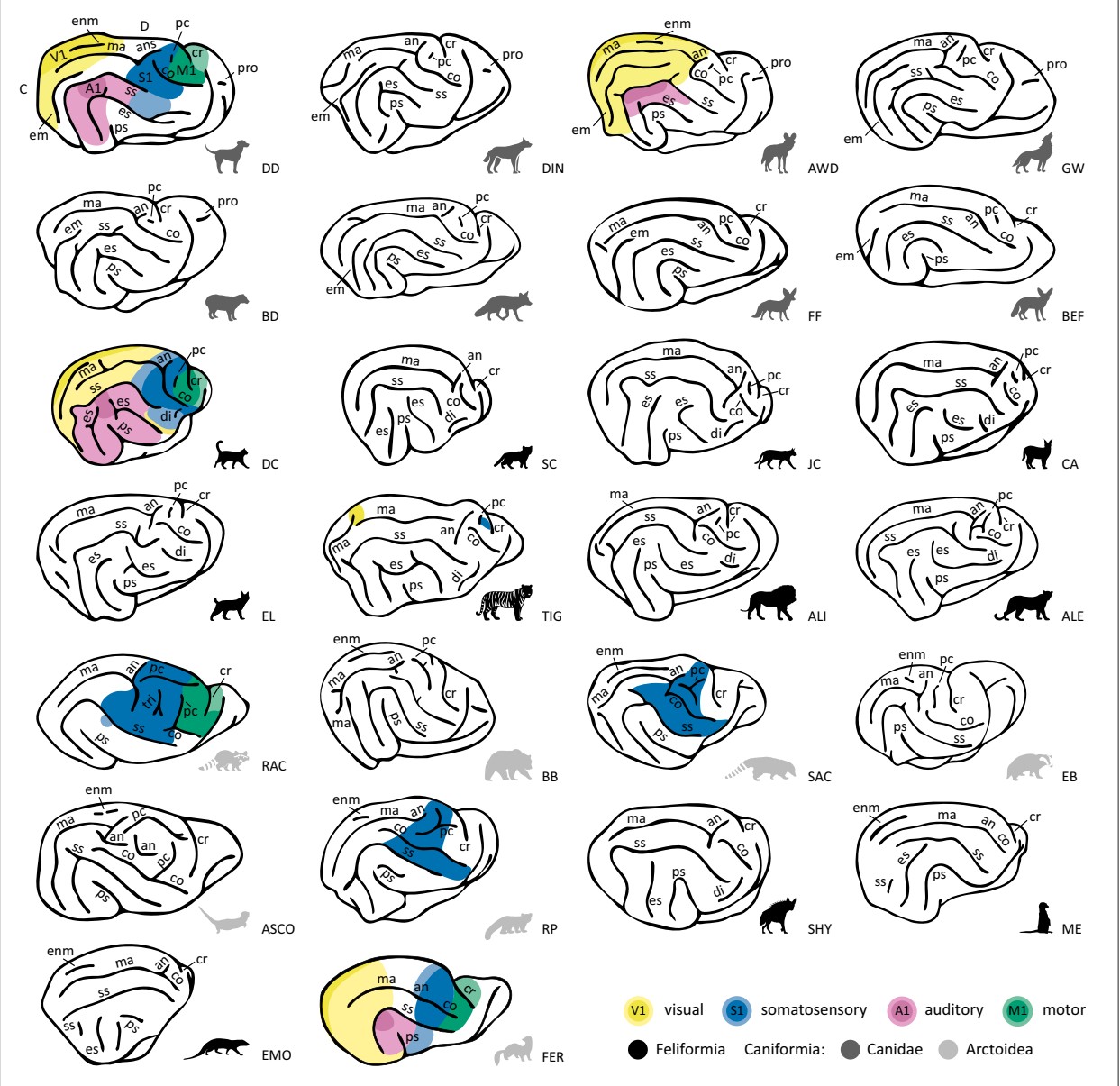

**Figure 6.** Schematic overview of cortical sulcal anatomy and available information about cortical sensory areas. While knowledge of the sensory regions in carnivoran brains is still limited, the best-understood brains of the species in our sample were the domestic cat (third row, left) and ferret (bottom row, right) followed by the domestic dog (top row, left). Based on prior electrophysiological, histological, and neuroimaging research (e.g., *Boch et al., 2021*; *Chengetanai et al., 2020a*; *Douglas Jameson et al., 1968*; *Guran et al., 2024*; *Hardin et al., 1968*; *Johnson et al., 2016*; *Kosmal, 2000*; *Law et al., 1988*; *Manger et al., 2002*; *Mclaughlin et al., 1998*; *Radtke-Schuller et al., 2020*; *Stolzberg et al., 2017*; *Tunturi, 1944*; *Welker and Campos, 1963*), we indicate approximate locations of unimodal sensory cortices on a lateral view of the brains of these species. Darker shades indicate primary sensory cortices, including the primary visual (V1, yellow), auditory (A1, pink), motor (M1), and somatosensory (S1) cortex. Lighter shades mark higher-order unimodal sensory regions. White spaces indicate that these regions have not been investigated yet, or research to date has revealed inconclusive results. It is, for example, possible that the African wild dog has additional auditory regions located ventral to those identified (*Chengetanai et al., 2020a*). C, caudal; D, dorsal. Sulcus acronyms: an, ansate; co, coronal; cr, cruciate; di, diagonal; em, ectomarginal; enm, endomarginal; es, ectosylvian; ma, marginal; pr, postcruciate; pro, proreal; ps, pseudo-sylvian; ss, suprasylvian; tri, triradiate. Animal acronyms: ALI, Asiatic lion; ALE, Amur leopard; ASCO, Asian small-clawed otter; AWD, African wild dog; BB, brown bear; BD, bush dog; BEF, bat-eared fox; CA, caracal; DC, domestic cat; DD, domestic dog; DIN, dingo; EB, Eurasian badger; EL, Eurasian lynx; EMO, Egyptian mongoose; FER, ferret; FF, fennec fox; GW, grey wolf; JC, jungle cat; ME, meerkat; RAC, raccoon; RF, red fox; RP, red panda; SAC, South American coati; SC, sand cat; SHY, striped hyaena; TIG, Bengal tiger.

endocasts in Canidae and Felidae observed the occipitotemporal cortex expansion taking place early in canid brain evolution and reported no observations of an ectomarginal sulcus in felids (*Lyras, 2009*; *Radinsky, 1969*). The meerkat and Egyptian mongoose had a unique sulcal configuration compared to all other species, with relatively less prominent suprasylvian and ectosylvian sulci forming an incomplete arc. The striped hyaena also stands out, exhibiting a complete suprasylvian arc but only a caudal ectosylvian sulcus.

Previous histological and electrophysiological investigations in the domestic cat (*Reale and Imig, 1980*), domestic dog (*Kosmal, 2000*; *Tunturi, 1950*), and African wild dog (*Chengetanai et al., 2020a*) indicate that large parts of the cortical territory between the middle and caudal parts of the suprasylvian and ectosylvian sulcus (i.e., on the ectosylvian gyrus) house primary and higher-order auditory regions (see also *Figure 6*). In the domestic cat, which has a split ectosylvian sulcus, the primary auditory cortex extends ventrally to occupy the cortical area between the sections of the split ectosylvian sulcus (*Reale and Imig, 1980*). It has been previously suggested that the expansion of the auditory cortex might have caused the split of the ectosylvian sulcus in Felidae brain evolution (*Radinsky, 1969*). Sensory regions beyond the somatosensory cortex remain unstudied in the majority of species without an ectosylvian sulcus in our sample; however, investigations in the ferret show that the auditory cortex is located ventral to the mid suprasylvian sulcus (*Bizley et al., 2005*). Thus, the suprasylvian sulcus appears to be a reliable marker of the location of unimodal auditory cortex in carnivorans.

Moreover, research on the domestic cat (*Reale and Imig, 1980*; *Stolzberg et al., 2017*) and domestic dog (*Kosmal, 2000*; *Tunturi, 1950*) indicates that the sylvian gyrus, which is only present in the species exhibiting an ectosylvian sulcus (see *Figure 4*), houses further higher-order auditory, but also visual and multisensory brain regions (see *Figure 6* for an overview of unimodal cortical regions). In the domestic dog, multisensory or visual cortical regions expand across the territory between the rostral ectosylvian sulcus and the pseudo-sylvian fissure (*Kosmal, 2000*; *Kosmal et al., 2004*), and in the domestic cat along the rostral ectosylvian sulcus (*Meredith et al., 2018*; *Stolzberg et al., 2017*). Furthermore, in both species, secondary somatosensory regions are bordering the rostral ectosylvian sulcus area. Thus, the rostral sylvian and ectosylvian cortical regions may represent a multisensory integration hub in felids and canids.

Another unique aspect of canid sulcal anatomy was the ectomarginal sulcus in the occipital lobe. Histological research on the African wild dog (*Chengetanai et al., 2020b*) and neuroimaging and diffusion MRI in domestic dogs (*Andrews et al., 2022*; *Boch et al., 2021*) shows that the ectomarginal gyrus and adjacent sulcal region are part of the extrastriate visual cortex, with the territory between the neighbouring marginal sulcus and dorsal convexity housing the primary visual cortex (V1; see *Figure 6*). The emergence of an ectomarginal sulcus may, therefore, indicate an expansion of the visual cortex in Canidae.

## Frontal cortical territory

A distinct proreal sulcus was observed in the frontal lobe of the domestic dog, the African wild dog, wolf, dingo, and bush dog. This may indicate an expansion of the frontal cortex in these animals compared to the other species in our sample (*Figures 5 and 6*). This aligns with findings from a comprehensive study comparing canid endocasts revealing an expanded proreal gyrus in these animals compared to the fennec fox, red fox, and other species of the genus Vulpes (*Lyras and Van Der Geer, 2003*). The canids with a proreal sulcus also exhibit complex social structures compared to the primarily solitary living foxes (*Nowak, 2005*; *Wilson and Mittermeier, 2009*; *Wilson, 2000* see *Figure 5*). Despite living in social groups, the bat-eared fox, an insectivorous canid, does not possess a proreal sulcus. Its foraging behaviour is best described as spatially or communally coordinated rather than truly cooperative (*Macdonald and Sillero-Zubiri, 2004*), suggesting that the relationship between sulcal morphology and sociality may be specific to species engaging in active cooperative hunting. Supplementary quantitative analyses also confirm an increase in the relative length of the proreal sulcus in cooperatively hunting species (see *Figure 5—figure supplement 1*, *Figure 5—source data 1*). Moreover, a previous investigation of Canidae and Felidae brain evolution, using endocasts of extant and extinct species, also suggested a link between the emergence of pack structures and the proreal sulcus in Canidae (*Radinsky, 1969*). Despite being highly social and living in large social groups (i.e., mobs), meerkats appear to have a relatively small frontal lobe and no proreal

sulcus compared to the social Canids (*Figure 5*), which would suggest that if the presence of a proreal sulcus correlates with complex social behaviour, this is canid specific.

## General discussion

Carnivorans represent a diverse order of mammals, characterised by a range of foraging and social behaviours, possessing relatively encephalised brains. In the present study, we explored cortical sulcal anatomy across a wide range of carnivoran species, marking the first in a series of communications dedicated to exploring carnivoran brain organisation. We provide a comprehensive overview of all major lateral and dorsal neocortical sulci in 26 carnivoran species and standardised recipes to identify each sulcus as a first reference frame to navigate and compare carnivoran brains. The recipes, derived from prior, partially incomplete descriptions and our own observations, provide a unified sulcal nomenclature and were designed to guide future explorations of lesser studied carnivoran brains. We then conducted a macro-level comparison of observed sulcal configurations across carnivoran species and families and found variations across lineages and species that may relate to how these animals interact with their environment for social structures and forage. Thus, our findings not only establish a framework to guide investigations into carnivoran brain organisation but also generate a large set of potential avenues to be addressed in future investigations.

Comparative analyses of encephalisation provide important insights into the evolutionary trajectory of carnivoran brains and potential factors influencing changes in relative brain size. Prior investigations found, for example, varying encephalisation shifts occurring independently in several families and at different rates (*Finarelli and Flynn, 2009*; *Michaud et al., 2022*). These changes in relative brain size were strongly correlated with home range size and geographic distribution in terrestrial Carnivorans (*Michaud et al., 2022*). However, although encephalisation is a useful measure to compare large numbers of species, its lack of specificity in terms of local expansions and other measures of brain organisation limits its ultimate use. By comparing sulcal anatomy, we observed lineage-specific regional variations in sulcal configuration potentially linked to the animal's behaviour and ecology, which cannot be detected by comparing relative brain sizes. Comparisons of regional brain size in social and solitary felids found, for example, a relationship between rostral cerebrum size and sociality in big cats (*Sakai et al., 2016*). Our work introduces another level, moving from whole-brain measurements to investigating patterns of sulcal configuration and potential functional correlates.

Our results revealed several interesting patterns of local variation in sulcal morphology between and within different lineages and successfully replicated and expanded upon prior observations based on more limited sets of species (*Radinsky, 1969*; *Radinsky, 1968*; *Welker and Campos, 1963*; *Welker and Seidenstein, 1959*). For example, Arctoidea showed relatively complex sulcal anatomy in the somatosensory cortex but low complexity in the occipitotemporal regions. In Canidae and Felidae, we found more complex occipitotemporal sulcal patterns indicative of changes in the amount of cortex devoted to visual and auditory processing in these regions. These observations may be linked to social or ecological factors, such as how the animals interact with objects or each other and their varied foraging strategies. Another example was the differential relative expansion of the neocortex surrounding the cruciate sulcus, which was particularly complex in Arctoidea species that are known to use their paws to manipulate their environment. Consistent with this observation, complementary quantitative analyses of both hemispheres revealed that species with high forepaw dexterity tended to have longer cruciate and postcruciate sulci. Although it has been argued that the cruciate sulcus appeared independently in different lineages and its exact relationship to the location of primary motor areas varies (*Radinsky, 1971*), our results provide a detailed exploration of the relationship between brain morphology and behavioural preferences across such a range of species.

Exploring the relationship between relative brain size and sulcal configuration, our findings showed that larger brains were more convoluted, but they did not consistently have the highest number of unique sulci. Despite not having the largest brains (*Michaud et al., 2022*), Canids exhibited the highest number of unique sulci, and the Bengal tiger and brown bear, with the largest brains in our sample, had the most convoluted brains with the deepest folds and numerous secondary sulci. Similarly, while the meerkat, among the smallest species in our study, exhibited a relatively smooth brain surface, its temporal cortex had a unique sulcal configuration. This included a caudal ectosylvian sulcus, which the meerkat shared exclusively with Felidae, Canidae, and the striped hyaena, but the sulcus was absent in

Arctoidea species, including the brown bear. These findings underscore that relative brain size alone does not account for the regional variations observed in carnivoran sulcal morphology.

Comparative neuroimaging requires balancing the level of anatomical detail with the breadth of species. The present sample represents the most comprehensive collection of fissiped carnivoran brains to date, encompassing a wide range of land-dwelling species from eight families. It includes diverse representatives, such as both social and solitary mongooses, weasel-like and non-weasel mustelids, and a broad array of canids, including wolf-like, fox-like, and more basal forms of canids. The framework and detailed protocols developed in this study are designed to facilitate navigation of additional fissiped species, such as Viverridae, Eupleridae, Mephitidae, Nandiniidae, and Priono-dontidae. Moreover, the approach can be readily extended to aquatic carnivorans, enabling broader macro-level comparisons across the order.

Regarding individual variability in external brain morphology, previous work in primates and carnivorans has shown that differences across individuals typically affect sulcal shape, depth, or extent, but not the presence of major sulci. This has been reported in diverse contexts, including comparisons between captive and (semi-)wild macaque (*Sallet et al., 2011*; *Testard et al., 2022*), different dog breeds (*Hecht et al., 2019*), domestic cats (*Kawamura, 1971*), or selectively bred foxes (*Hecht et al., 2021*). By including additional individuals for selected species, we extend these findings to a broader range of carnivorans. Notably, we observed no major sulcal differences between closely related species, even when specimens were acquired using different extraction and scanning protocols, for example, across felid clades or among wolf-like canids, further suggesting that substantial within-species variation is unlikely. While a full analysis of interindividual variability lies beyond the scope of this study, our findings support the reliability of the major sulcal patterns described.

We focused in this work on cortical surfaces reconstructed from whole-brain MRI scans and created detailed digital atlases of the identified sulci. This contrasts with the more traditional approach of labelling photographs of the samples. There are a number of reasons why labelling MRI scans can be advantageous. The first is that it allows a direct integration with other imaging modalities acquired from the same brains. For example, most of the data reported here were derived from diffusion MRI scans, which will allow us to perform reconstructions of the major white matter bundles in these species (e.g., *Jacqmot et al., 2013*). Second, the digital nature of MRI data allows more advanced, quantitative comparisons, both within and across species (cf. *Mars et al., 2021*). For instance, reconstruction of major white matter bundles based on diffusion MRI data has been shown to be a useful measure of comparison across species in primate research (cf. *Mars et al., 2018*; *Warrington et al., 2022*) and comparison across different modalities has been shown to be an insightful way to distinguish competing hypotheses of evolutionary change across lineages, such as whether differences are due to local expansion of homologous areas or changes in the connectivity of areas (*Eichert et al., 2020*; *Eichert et al., 2019*). Third, neuroimaging data are not limited to 2D representations and can be easily shared between researchers, allowing data of rare species and samples to be studied by more groups and using distinct approaches. We have created the Digital Brain Zoo (*Tendler et al., 2022*) for this specific purpose and the data underlying this project is deposited there.

The sulcal anatomy provided here will equip future work with a reliable reference frame, and the observed functional correlates with sulcal morphology provide distinct hypotheses to test. Specifically, the sulcal morphology diversity in the occipitotemporal cortex is intriguing, in particular with respect to the known anatomical variation of the temporal cortex in primates (*Braunsdorf et al., 2021*), the homology of which across orders remains to be established (*Bryant and Preuss, 2018*). Formal comparisons in local brain organisation can also aid in the further integration of information across levels. For instance, it is known that the dog temporal lobe houses a number of areas specific to social information processing (*Andics and Miklósi, 2018*; *Bálint et al., 2023*; *Boch et al., 2024a*; *Boch et al., 2024b*; *Boch et al., 2023*). This dovetails with our observation of the complex sulcal organisation of this part of the brain in more social species (*Figure 5B*). Previous comparative work on the carnivoran social brain has highlighted the extended proreal cortex in social species (*Holekamp et al., 2007*; *Radinsky, 1969*). This is a finding we replicate both qualitatively and quantitatively through the presence and increased length of the proreal sulcus in these animals, but it also shows how focusing purely on measures of brain encephalisation reveals only a limited picture. A similar observation has been made in primates, where larger brains tend to correlate with larger social groups (*Dunbar and Shultz, 2007*), but this finding does not describe the complexity of the homology of brain regions

processing social information across species (*Mars et al., 2013*; *Roumazeilles et al., 2021*; *Wittmann et al., 2018*). As another case in point, the meerkat, a highly social species, does not have a proreal sulcus but does have an expanded proreal gyrus compared to the less social mongoose species (*Radinsky, 1975b*). Similarly, the anterior cerebrum (i.e., endocast volume anterior to cruciate sulcus) was most expanded in the highly social spotted hyaenas compared to its more solitary evolutionary relatives, such as the striped hyaena (*Sakai et al., 2011*).

In summary, our study transcends traditional brain size comparisons, illustrating the diversity of carnivoran brain organisation and regional differences. We propose potential links between the observed variation in sulcal anatomy and the species' behavioural ecology, and by establishing stan-dardised criteria, we lay the groundwork for a better understanding of lesser studied carnivoran brains. Finally, we provide the roadmap for further anatomical investigations exploiting the use of neuroimaging in the comparative study of carnivoran and mammalian brain diversity.

## Limitations and future directions

Our findings represent a critical first step for linking brains within and across species for interspecies insights. The present analyses are based on multiple individuals pooled into families and genera, primarily focusing on single representatives per species. Additional individuals for selected species confirmed that intra-species variation is a matter of degree rather than a case of presence or absence of major sulci, but we do not provide an extensive account of the possible range of sulcal shape or other anatomical features.

Future studies will aim to systematically investigate interindividual variability in sulcal shape, depth, surface area, or thickness of the cortical ribbon surrounding the sulci, and will extend to more detailed investigations of the medial part of the cortex, as well as the subcortical structures and the cere-bellum. The present framework and resulting database also provides the foundation to guide and facilitate future investigations of inter- and intra-species variation in regional brain size.

# Materials and methods

We obtained the diverse carnivoran brain data mainly through post-mortem samples. Due to varia-tions in scanning protocols (*Table 1*), we homogenised the data set during the initial preprocessing step. Following that, we employed a standardised pipeline to generate cortical surfaces for all brains. We then used the neocortical surface reconstructions to systematically label all major neocortical sulci in each brain. To guide the labelling process, we created standardised criteria (recipes) containing detailed descriptions of how to identify each sulcus based on available prior descriptions and our own observations.

## Data

The sample consists of 26 carnivoran species (*Figure 1*). All data, except for the cat, were acquired ex vivo (see *Table 1* for detailed sample descriptives). Analyses reported in the main text are all based on a single, adult or subadult individual of each species. We show confirmatory analyses on addi-tional samples for five species and an averaged template for the domestic dog in the supplementary material.

## Procedure

Brains of all post-mortem samples were processed within 24 hr of death. The majority of the samples were obtained by the Copenhagen Zoo and the Zoological Society of London (see *Table 1* for details). The Copenhagen Zoo specimens were all collected immediately after euthanasia. Following general anaesthesia and an overdose of sodium pentobarbital (200 mg/kg, i.v.), the heads of animals were perfusion-fixed post-mortem through the carotid arteries, initially with a rinse of 0.9% saline followed by 4% paraformaldehyde (PFA) in 0.1 M phosphate buffer. The brains were then removed from the skull and post-fixed in PFA for 24 hr and subsequently transferred to a sodium azide phosphate buffer solution to ensure preservation until MRI data acquisition. Antemortem observations for all these animals revealed that they were in good health with no obvious neural deficits or behavioural abnor-malities, but were euthanised for population management reasons (*Bertelsen, 2019*). Samples from the Zoological Society of London's pathology archive were all from captive animals, with the exception

of the Eurasian badger, which was a wild animal found dead on site. The brains had been removed as part of routine post-mortem examinations and fixed and stored in 10% neutral buffered formalin. No evidence of brain pathology was noticed during the necropsy.

The raccoon and Ussuri brown bear were euthanised as part of pest control measures by the city of Iwamizawa (Hokkaido, Japan), a decision unrelated to the current study. After extraction, the brains were immersed in 4% PFA for 5 days and subsequently transferred to a sodium azide phosphate buffer solution to ensure preservation until MRI data acquisition.

The fennec and red fox brain specimens were removed within 14 hr of death and immersion fixed in 10% formalin at necropsy before being transferred to a solution of 0.1 M phosphate-buffered saline (PBS) with 0.1% sodium azide solution and stored at 4°C prior to scanning. The fennec fox specimen was raised in captivity at the St. Louis Zoo before being donated to M.A.S., while the red fox was donated to M.A.S. by a local taxidermist.

Samples from the Mammalian MRI (MaMi) database (*Assaf et al., 2020*) were collected following the incidental death of animals in zoos in Israel or natural death collected abroad, and with the permission of the national park authority (approval no. 2012/38645) or its equivalent in the relevant countries. No animals were deliberately euthanised for this study. All brains were extracted within 24 hr of death and fixed in 10% formaldehyde for several days to a few weeks, depending on brain size. Prior to MRI scanning, the brains were rehydrated in PBS for approximately 24 hr. To minimise image artefacts caused by magnetic susceptibility effects, the brains were immersed in fluorinated oil (Fluorinert, 3 M) inside a plastic bag during scanning.

For the bat-eared fox and sand cat samples from the Lyon Comparative Brain Collection, the heads were immersed in 10% formalin solution within 12 hr of death. The brains were then removed from the skulls within the next 72 hr and placed in formalin solution. After at least 4 weeks, the brain was placed in a PBS solution for a minimum of 3 days before being transferred into a fluorinert solution for scanning purposes. To eliminate trapped air bubbles, samples were subjected to vacuum pumping prior to scanning. Following the scans, the samples were returned to formalin for storage.

To obtain in vivo data of the domestic cat, a board-certified anaesthesiologist performed general anaesthesia on the animal. It was premedicated with dexmedetomidine (10 mg/kg; Zoetis Inc, Kalamazoo, MI), induced using propofol for general anaesthesia and dosed to effect (10–20 mg/kg; Sagent Pharmaceuticals, Schaumburg, Ill), and intubated. The animal was maintained under anaesthesia using oxygen and inhalant isoflurane and supportive intravenous lactated Ringer's solution fluids.

## Imaging protocols and data preprocessing

Data was obtained using 3 and 7T wide-bore human MRI scanners or 7T narrow-bore non-human animal scanners (see *Table 1*). Scanning protocols included post-mortem T2-weighted and in vivo T1-weighted structural or post-mortem diffusion protocols. In all cases, we preprocessed the data using tools from FSL (https://fsl.fmrib.ox.ac.uk/fsl/docs/), ANTs (*Avants et al., 2009*), and custom code from the Comparative Anatomy Toolbox (Mr Cat; http://www.neuroecologylab.org/) to create scans with a T1-like contrast needed for surface reconstruction.

Large post-mortem samples from the Copenhagen Zoo specimen collection and the Zoological Society of London (see *Table 1*) were scanned at the University of Oxford on a wide-bore human 7T scanner with 70 mT/m maximum gradient. Diffusion MRI data were acquired using a diffusion-weighted steady-state free-precession (DW-SSFP) sequence ($q$-value = 300 cm$^{-1}$, gradient duration = 13.56 ms, gradient amplitude = 52 mT m$^{-1}$, flip angle = 39°, 10–13 non-diffusion-weighted volumes per brain, TE/TR = 21/29 ms, EPI factor = 1, Bandwidth = 100 Hz per pixel) at 600 µm isotropic resolution with 160 diffusion directions, using a 1 Tx/28 Rx QED knee coil (European wolf, Asiatic lion, brown bear) or 1 Tx/32 Rx Nova head coil (American wolf, amur leopard, African wild dog). The acquisition took 16 min and 25 s per volume. Datasets were corrected for Gibbs ringing (*Kellner et al., 2016*) and co-registered using FSL FLIRT (*Jenkinson and Smith, 2001*). Diffusion Tensor and Ball & 2 Sticks model estimates were derived using custom software accounting for the full DW-SSFP signal model (*Buxton, 1993*), implemented using cuDIMOT (*Hernandez-Fernandez et al., 2019*). To account for the dependencies of DW-SSFP on relaxation times and flip angle, quantitative $T_1$, $T_2$, and $B_1$ maps were additionally estimated in each sample using a turbo inversion-recovery, turbo spin-echo and actual flip angle imaging sequence (*Yarnykh, 2007*), respectively. A ball and two stick

model was fitted to the data using a modified version of bedpostX (*Behrens et al., 2007*). Diffusion modelling code is available at https://github.com/BenjaminTendler/DW-SSFP, copy archived at *Tendler, 2023*.

Small post-mortem samples from the Copenhagen Zoo specimen collection, the London Zoological Society, and the domestic dog were scanned at the University of Oxford using a narrow-bore rodent 7T MRI scanner (Varian, Oxford UK) with 400 mT/m maximum gradient. Acquisition used a 2D diffusion-weighted spin-echo multi-slice protocol with single line readout DW-SEMS; TR/TE = 19/26 ms; matrix size = 128 × 128 with a sufficient number of slices to cover each brain; spatial resolution for the dingo, domestic dog (Belgian shepherd), and lynx was.6 mm isotropic, for the bush dog, red panda, and coati: 0.5 mm isotropic, for the Eurasian badger 0.4 mm isotropic, and 0.3 mm for the meerkat. A total of 16 non-diffusion-weighted ($b = 0$ s/mm$^2$) and 128 diffusion-weighted ($b = 4000$ s/mm$^2$) volumes were acquired with diffusion encoding directions evenly distributed over the whole sphere (single shell protocol). All data were preprocessed using the same protocol implemented in the module phoenix of the MR Comparative Anatomy Toolbox (Mr Cat; http://www.neuroecologylab.org). Briefly, the steps are as follows: We first converted the datasets to NIFTI format, then built an image based on the volumes acquired without a diffusion gradient as well as a binary mask of this image. Diffusion Tensor and Ball & 3 Sticks model estimates were derived using dtifit and BedpostX tools from FSL (https://fsl.fmrib.ox.ac.uk/fsl/docs/).

Structural brain scans of the raccoon and Ussuri brown bear were acquired at the Medical School of Hokkaido University using a wide-bore 3T human MRI scanner with a 16-channel body coil. Imaging was performed using a 3D turbo spin-echo sequence, with 172 and 320 slices for the raccoon and bear, respectively. Isotropic spatial resolution was 0.63 mm for the raccoon and 0.78 mm for the Ussuri brown bear, with echo train lengths of 179 and 246. For the raccoon, additional imaging parameters were as follows: TR/TE = 4000/411 ms, FoV = 107 × 120 mm, matrix 192 × 172, flip angle = 120°, and a bandwidth of 400 Hz per pixel. The scan lasted approximately 10 min and 56 s. For the Ussuri brown bear, parameters included TR/TE = 4000/410 ms, FoV = 237 × 200 mm, matrix size = 304 × 256, flip angle = 120°, and a bandwidth of 700 Hz/pixel. The scan lasted approximately 12 min and 24 s.

The in vivo structural scan of the domestic cat was performed at Cornell Magnetic Resonance Imaging Facility (CMRIF) using a wide-bore human 3T (General Electric Discovery MR750), with 50 mT/m maximum gradient strength. The animal was placed with the head centred and in dorsal recumbency in a 16-channel small flex radiofrequency coil (Neocoil, Pewaukee, WI NeoCoil). Data was acquired using a T1-weighted 3D MPRAGE (BRAVO) sequence with 0.5 mm$^3$ isotropic voxel resolution and TR/TE of 8.436/3.604 (TI = 450, flip angle = 12°, NEX = 1). The scan lasted approximately 6 min and 30 s. Before surface creation, the scan was intensity bias-corrected using ANTs (*Avants et al., 2009*).

Scanning of the red and fennec fox was undertaken at the Icahn School of Medicine (Mt. Sinai, NY) using a narrow-bore rodent 7T Bruker Biospec scanner. A 3D FLASH (fast low angle shot) sequence was used with the following parameter settings: TR/TE = 36/23 ms, flip angle = 15°, FoV = 128 × 128× 175, matrix size = 384 × 384× 384 mm in each slab with a spatial resolution of 0.13 mm isotropic for the fennec fox and 0.18 mm isotropic for the red fox.

In the MaMi database (*Assaf et al., 2020*), smaller brains were scanned using a 7T 30/70 BioSpec Avance Bruker system, while larger brains were scanned using a 3T Siemens Prisma system (see *Table 1*). High-resolution structural scans (T2- or T1-weighted MRI) were acquired as anatomical references for diffusion-weighted imaging, using a fixed matrix size of 128 × 96 and spatial resolution scaled to brain size. The following isotropic voxel sizes were used for the brains included in this study: caracal (0.25 mm$^3$), ferret (0.3 mm$^3$), Persian leopard (0.3 mm$^3$), Egyptian mongoose (0.34 mm$^3$), sand cat (0.37 mm$^3$), second red panda (0.41 mm$^3$), jungle cat (0.48 mm$^3$), striped hyaena (0.62 mm$^3$), and Bengal tiger (0.7 mm$^3$).

T2-weighted structural scans from the Lyon Comparative Brain Collection (see *Table 1*) were acquired with a 3T Siemens Prisma system and a 3D turbo spin-echo SPACE sequence. Scans consisted of 288 slices with an isotropic voxel resolution of 0.3 mm$^3$ for the bat-eared fox and 0.25 mm$^3$ for the sand cat. Additional parameters included: TR/TE = 3000/26 ms, FoV = 128 × 128, matrix size = 512 × 512, and a bandwidth of 166 Hz per pixel.

## Surface creation

All scans were reoriented to anterior/posterior commissure and standard FSL orientation. We then generated cortical surface meshes using *precon_all* (*Benn et al., 2025*; https://github.com/neura-benn/precon_all; *Benn, 2025*) an adapted version of Freesurfer's recon-all pipeline (*Fischl, 2012*), designed to create surfaces of non-human animal models.

*Precon_all* reconstructs the surfaces based on scans with a T1-like contrast. After intensity bias-field correction using ANTs, we, therefore, inverted the voxel intensities of the T2-weighted scans (red fox, fennec fox, raccoon, bat-eared fox, both sand cats, ferret, Ussuri brown bear, Jungle cat, Egyptian mongoose, and striped hyaena) by multiplying them with –1 and keeping cerebrospinal fluid (csf) intensities at zero. *Precon_all* requires a spatial resolution of 0.2 mm or higher. We, therefore, upsampled the scans to 0.5 mm$^3$ for the red fox and 0.3 mm$^3$ for the caracal and the fennec and bat-eared fox using FSL FLIRT (*Jenkinson and Smith, 2001*) to strike a balance between preserving information and achieving optimal surface generation. For the diffusion MRI scans, we created the T1-like images by calculating the square root of the sum of the mean_f{1,2,3}_samples (indicating the mean of the anisotropy distribution at each voxel).

Next, we removed any remaining non-brain tissue from all brain scans using individual brain masks, which we created using FSL's *bet* or ITK-snap's segmentation tool (*Yushkevich et al., 2006*) and, if necessary, manually improved them. We then created masks for each hemisphere using FSLeyes and drew 'subcortical' and 'non-cortical' masks in ITK-snap (see *Benn et al., 2025* for a detailed description of the pipeline). The subcortical masks comprised the corpus callosum expanding to the outer borders of the lateral ventricles, which are filled during surface generation. We included the cerebellum, brain stem, and olfactory bulb in the non-cortical mask to remove them from the surfaces. Following an initial run of *precon_all*, surfaces were refined by manually editing the resulting white matter masks. We then ran *precon_all* again to generate white, pial, and mid-thickness surfaces and down-sampled them to 10,000 vertices and applied spatial smoothing using connectome workbench tools (*Marcus et al., 2011*) to facilitate sulcal labelling. For the labelling of lateral, ventral, and medial wall sulci, we used a smoothing strength of 0.5, and to detect partially more shallow dorsal sulci, we labelled surfaces with a 0.2 smoothing strength (15 iterations each).

## Labelling, creation of recipes, and sulcal masks

We focused on the major lateral and dorsal sulci of the carnivoran brain, but the medial wall and ventral view of the sulci are also described. Although any definition of 'major sulci' is to some extent arbitrary, we here use the term to refer to those sulci that are particularly prominent in the majority of species and can be unambiguously defined. For consistency, we started by labelling the right hemispheres on the mid-thickness surfaces; these are the hemispheres presented in the manuscript. An exception was made for the jungle cat, for which only the left hemisphere was available and is therefore shown. We aimed to facilitate interspecies comparisons and the exploration of previously undescribed carnivoran brains. To this end, we first created standardised criteria (henceforth referred to as recipes) for identifying each sulcus, drawing from existing literature on carnivoran neuroanatomy, particularly in paleoneurology (*Lyras et al., 2023*), and our own observations. In addition, we created digital sulcal masks for both hemispheres, which allowed us to test whether the same patterns were observable bilaterally and to further facilitate future research building on our framework. For the Egyptian mongoose, only the right hemisphere was available, and thus, a bilateral comparison was not possible for this species. Anatomical nomenclature primarily follows the recommendations of *Czeibert et al., 2018*; if applicable, alternative names of sulci are provided once.

We started creating the recipes based on felidae and canidae neuroanatomy descriptions since these species' brains' outer morphology and evolutionary history are the most extensively described within the order carnivora (see e.g., *Chengetanai et al., 2020c*; *Lyras, 2009*; *Radinsky, 1973b*; *Radinsky, 1975a*; *Radinsky, 1969*; *Rogers Flattery et al., 2023*; *Sakai et al., 2016*). We then completed the recipes with the observations made while labelling the other carnivoran brains and, if available, confirmed them with prior neuroanatomical descriptions. The ferret brain has also been described in detail (*Radtke-Schuller, 2018*), and relatively extensive neuroanatomical descriptions are available for members of the Ursidae and Hyaenidae families (e.g., *Sakai et al., 2011*; *Sienkiewicz et al., 2019*; *Smith, 1933*). However, for most other carnivoran species, only a few and often partial descriptions exist (but see *England, 1973* for comparisons of a wide variety of species). Prior

descriptions of the Asian small-clawed otter, South American coati, raccoon, and red panda brain primarily focus on the somatosensory and motor cortex (*Hardin et al., 1968*; *Johnson et al., 1982*; *Radinsky, 1968*; *Welker and Campos, 1963*; *Welker and Seidenstein, 1959*). While descriptions and schematic drawings of the Eurasian badger and meerkat exist, not all sulci are labelled and described in detail, as the investigations focused on other aspects (*Radinsky, 1975b*; *Radinsky, 1973a*). If not mentioned otherwise, our observations were in line with available prior descriptions of these species' neuroanatomy.

## Sulcal anatomy variations and potential relationship to function

We first briefly illustrated the gyri of the carnivoran brain with a focus on gyri that are not present in some species as a consequence of absent sulci to complement our observations. We then summarised the key differences and similarities in sulcal anatomy between species and related them to their ecology and behaviour. To complement this qualitative description, we conducted an initial quantitative analysis of sulcal length data from both hemispheres.

To test whether sulcal length covaries with behavioural traits, we fit linear models predicting the relative length of the three target sulci (cruciate, postcruciate, proreal) as a function of forepaw dexterity (low vs. high) and sociality (solitary vs. cooperative hunting). We measured the absolute length of each sulcus using the *wb_command -border-length* function from the Connectome Workbench toolkit (*Marcus et al., 2011*) applied to the manually defined sulcal masks (i.e., border files). Relative sulcal length was calculated by dividing the length of each target sulcus by that of a reference sulcus in the same hemisphere, reducing interspecies variation in brain or sulcal size. Reference sulci were required to be present in all species within a hemisphere and excluded if they were a target sulcus, part of the same functional system (e.g., somatosensory/motor), or anatomically atypical (e.g., the pseudo-sylvian fissure). This resulted in seven reference sulci for the proreal sulcus (ansate, coronal, marginal, presylvian, retrosplenial, splenial, and suprasylvian) and four for the cruciate and postcruciate sulci (marginal, retrosplenial, splenial, and suprasylvian). For each target–reference pair, we fit the following linear model: relative length ~ forepaw dexterity + sociality. Models were run separately for left and right hemispheres, with the left serving as a replication test. Associations were considered meaningful if the predictor reached statistical significance ($p \leq 0.05$) in ≥75% of reference sulcus models per hemisphere. Additional individuals were not included in the analysis.

Finally, we created schematic drawings of the observed sulcal patterns, and, if known, we also denoted approximate locations of sensory regions to add another layer of comparison that allows for indications of cortical areas that potentially expanded. We obtained information about sensory properties from prior histological and electrophysiological research. For the majority of the species, functional properties of brain areas have not been studied yet, or the focus was only on the somatosensory cortex, such as for the South American coati and red panda (*Welker and Campos, 1963*). The sensory cortices of the domestic cat are best understood and studied most extensively (see e.g., *Stolzberg et al., 2017* for a detailed atlas of the cat brain), followed by the ferret (e.g., *Law et al., 1988*; *Manger et al., 2002*; *Mclaughlin et al., 1998*; *Radtke-Schuller et al., 2020*), domestic dog (*Adrian, 1941*; *Kosmal, 2000*; *Kosmal et al., 2004*; *Bromiley et al., 1956*; *Tunturi, 1944*), raccoon (*Douglas Jameson et al., 1968*; *Hardin et al., 1968*; *Welker and Seidenstein, 1959*), African wild dog (*Chengetanai et al., 2020a*; *Chengetanai et al., 2020b*), and tiger (*Johnson et al., 2016*).

## Acknowledgements

The authors would like to thank Yaniv Assaf for providing access to the Mammalian MRI (MaMI) database (*Assaf et al., 2020*); Boras Zoo, Parken Zoo, Randers Regnskov, and Ree Park for contributing samples to the Copenhagen Zoo specimen collection and João Paulo Coimbra for his help collecting the data. We also thank Franck Lamberton and Céline François-Brazier for their help in acquiring the data for the Lyon Comparative Brain Collection, and Katharina Langer for her assistance in creating the digital sulcus masks. For the purpose of Open Access, the author has applied a CC BY public copyright licence to any Author Accepted Manuscript version arising from this submission. MB received support for this project through the Postdoc Award granted by the Faculty of Psychology, University of Vienna; MB is a recipient of the L'ORÉAL Austria fellowship within the initiative 'For Women in Science' and her research was funded in part by the Austrian Science Fund (FWF) 10.55776 / J4828. The work of RBM is supported by the Biotechnology and Biological Sciences Research Council

(BBSRC) UK [BB/N019814/1] and the Medical Research Council (MRC) UK [MR/Y010698/1]. CL acknowledges partial funding for the research described herein from the Austrian Science Fund (FWF) 10.55776/P34675. MAS received support from the Verizon Foundation. BCT is supported by a Sir Henry Wellcome Postdoctoral Fellowship (Wellcome Trust) [222829/Z/21/Z]. KLM is supported by the Wellcome Trust [202788/Z/16/Z, 224573/Z/21/Z]. The Wellcome Centre for Integrative Neuroimaging (now called Oxford Centre for Integrative Neuroimaging) is supported by core funding from the Wellcome Trust [203139/Z/16/Z]. The funders had no role in study design, data collection and analysis, decision to publish, or preparation of the manuscript. For the purpose of Open Access, the authors have applied a CC BY public copyright licence to any Author Accepted Manuscript version arising from this submission.

## Additional information

### Funding

| Funder | Grant reference number | Author |
| --- | --- | --- |
| Austrian Science Fund | 10.55776/J4828 | Magdalena Boch |
| Postdoc Award, Faculty of Psychology, University of Vienna | | Magdalena Boch |
| For Women in Science L'OREAL UNESCO Award | | Magdalena Boch |
| Biotechnology and Biological Sciences Research Council | BB/N019814/1 | Rogier B Mars |
| Austrian Science Fund | 10.55776/P34675 | Claus Lamm |
| Verizon Foundation | | Muhammad A Spocter |
| Wellcome Trust | 10.35802/222829 | Benjamin C Tendler |
| Wellcome Trust | 10.35802/202788 | Karla L Miller |
| Wellcome Trust | 10.35802/224573 | Karla L Miller |
| Medical Research Council | MR/Y010698/1 | Rogier B Mars |

The funders had no role in study design, data collection, and interpretation, or the decision to submit the work for publication. For the purpose of Open Access, the authors have applied a CC BY public copyright license to any Author Accepted Manuscript version arising from this submission.

### Author contributions

Magdalena Boch, Conceptualization, Data curation, Formal analysis, Investigation, Visualization, Methodology, Writing – original draft, Project administration, Writing – review and editing; Katrin Karadachka, Kep-Kee Loh, Lea Roumazeilles, Methodology, Writing – review and editing; R Austin Benn, Software, Methodology, Writing – review and editing; Mads F Bertelsen, Ethan Wriggelsworth, Simon Spiro, Philippa J Johnson, Kamilla Avelino-de-Souza, Nina Patzke, Claus Lamm, Karla L Miller, Jérôme Sallet, Alexandre A Khrapitchev, Resources, Writing – review and editing; Paul R Manger, Muhammad A Spocter, Resources, Validation, Writing – review and editing; Benjamin C Tendler, Resources, Methodology, Writing – review and editing; Rogier B Mars, Conceptualization, Resources, Data curation, Supervision, Investigation, Methodology, Writing – original draft, Writing – review and editing

### Author ORCIDs

Magdalena Boch  https://orcid.org/0000-0003-3627-5180
R Austin Benn  https://orcid.org/0000-0003-3728-7036
Mads F Bertelsen  https://orcid.org/0000-0001-9201-7499
Claus Lamm  https://orcid.org/0000-0002-5422-0653
Karla L Miller  https://orcid.org/0000-0002-2511-3189

Alexandre A Khrapitchev https://orcid.org/0000-0002-7616-6635
Benjamin C Tendler https://orcid.org/0000-0003-2095-8665
Rogier B Mars https://orcid.org/0000-0001-6302-8631

### Ethics

The in vivo data collection of the domestic cat was approved by the Institutional Animal Care and Use Committee at Cornell University (protocol number 2015-0115); and a board-certified veterinary anaesthesiologist (American College of Veterinary Anesthesia) approved all protocols. All ex vivo data was obtained from animals euthanised or deceased for reasons unrelated to this study.

Reviewer #1 (Public review): https://doi.org/10.7554/eLife.100851.3.sa1
Author response https://doi.org/10.7554/eLife.100851.3.sa2

## Additional files

### Supplementary files

MDAR checklist

### Data availability

All derived data supporting the article, including sulcal masks, associated midthickness cortical surface reconstructions for all 32 animals, species-specific behavioural data, and the code used to extract sulcal lengths and perform the statistical analyses, are publicly available at https://git.fmrib.ox.ac.uk/neuroecologylab/carnivore-surfaces (copy archived at *Boch, 2025*). Sulcal length data were created, extracted, and analysed using the Connectome Workbench toolkit (*Marcus et al., 2011*), R version 4.4.0 (R Core Team, 2023), and Python version 3.9.7. The corresponding raw data (T1-like volumetric MRI scans) are available with access restrictions. Raw data for the majority of species are available via the Digital Brain Zoo of the University of Oxford (*Tendler et al., 2022*; https://open.win.ox.ac.uk/DigitalBrainBank/#/datasets/zoo), subject to agreement to a Material Transfer Agreement in accordance with ethical and funding requirements. Exceptions are raw data obtained from the Mammalian MRI (MaMI) database (*Assaf et al., 2020*), which are available directly from that database under its own access conditions (see Table 1 for an overview of data sources); and the in-vivo data of the domestic cat, available upon request from co-author P.J.J. under similar access procedures.

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

## Appendix 1

### Supplementary Note 1: Major sulci from ventral perspective

The majority of the sulci seen from a ventral view (*Figure 2—figure supplement 1*) were already described in the main text, as they were visible from lateral and dorsal views (see *Figures 2 and 3*).

The lateral rhinal fissure represents the ventral border of the neocortex in carnivorans. The pseudo-sylvian fissure originates from the lateral rhinal fissure, dividing it into a caudal (light green; *Figure 2—figure supplement 1*) and rostral (light blue; *Figure 2—figure supplement 1*) part. The sulcal anatomy of the brown bear stood out compared to the other species as the caudal portion of the caudal lateral rhinal fissure merged with the retrosplenial sulcus (*Figure 2—figure supplement 1*, yellow; and see Appendix—Supplementary Note 2), and the marginal sulcus (*Figure 2—figure supplement 1*, cyan) extended to the ventral surface terminating closely to the rostral end of the caudal lateral rhinal fissure. In all other species, the retrosplenial did not merge with the caudal lateral rhinal fissure, and the marginal sulcus was not visible on the ventral surface as it terminated more dorsally.

The ventral lateromedial sulcus (*Figure 2—figure supplement 1*, pink) is only visible from a ventral perspective and is orthogonal to the occipitotemporal (red) and retrosplenial sulcus (yellow; see also Appendix—Supplementary Note 2) positioned between the two sulci and the rostral lateral rhinal fissure (*Figure 2—figure supplement 1*, light green). We were able to identify the ventral lateromedial sulcus in the Asian lion, brown and Asian small-clawed otter; and in most Canidae, except for the domestic dog and fennec fox. This sulcus has been previously described as not consistently identifiable in domestic dogs (*Czeibert et al., 2018*).

### Supplementary Note 2: Major sulci of the medial wall

The cruciate sulcus (*Figure 3—figure supplement 1*, light blue) was also visible on the medial wall and merged in the majority of the species with the dorsal longitudinal splenial sulcus (*Figure 3—figure supplement 1*, purple red). The splenial sulcus runs caudally and curves ventrally at the caudal end. The portion of the sulcus after the bend is called the retrosplenial sulcus (*Figure 3—figure supplement 1*, *Figure 3—figure supplement 2*, yellow). In most of the brains in the present sample, the cruciate, splenial and retrosplenial sulci merged, or two of the three sulci merged; only in the brown bear and Ussuri brown bear were all three sulci detached.

In the wolf-like Canidae (the domestic dogs, dingo, African wild dog, and grey wolves), the Eurasian lynx, sand cats, Bengal tiger, Asiatic lion, striped hyaena and both the Amur and Persian leopard, we noted another sulcus curving around the caudal bend of the retro-/splenial sulcus. The dorsal portion of the sulcus is called the suprasplenial sulcus (*Figure 3—figure supplement 1*, *Figure 3—figure supplement 2*, green), and the ventral portion is the occipitotemporal sulcus (*Figure 3—figure supplement 1*, purple). In the dingo, Bengal tiger, Asiatic lion, striped hyaena, and the Amur and Persian leopard, the suprasplenial and the occipitotemporal sulcus were detached. In most other species, only one of the two sulci could be identified, and in the domestic cat, fennec fox, South American coati, and ferret, neither was observed.

