## [Editor Report · eLife Assessment]

This **valuable** study presents the first detailed and comprehensive description of brain sulcus anatomy of a range of carnivoran species based on a robust manual labeling model allowing species comparisons. The database and method for reconstructing cortical surfaces are **compelling**, and the evidence supporting the conclusions is **solid**. Despite the additional specimen, the evaluation of intra-species variations remains limited, but an insight into the inter-individual variability is now available for certain species. Exploring the associations between sulcal length and behavioral characteristics further suggests the potential of sulci as a proxy of functional organization. Setting an instructive foundation for comparative anatomy, this study will be of interest to neuroscientists and neuroimaging researchers interested in that field, as well as in brain morphology and sulcal patterns, their phylogeny and ontogeny in relation to functional development and behaviour.

---

## [Referee Report · Reviewer #1 (Public review)]

Summary:

This paper by Boch and colleagues, entitled Comparative Neuroimaging of the Carnivore Brain: Neocortical Sulcal Anatomy, compares and describes the cortical sulci of eighteen carnivore species, and sets a benchmark for future work on comparative brains.

Based on previous observations, electrophysiological, histological and neuroimaging studies and their own observations, the authors establish a correspondence between the cortical sulci and gyri of these species. The different folding patterns of all brain regions are detailed, put into perspective in relation to their phylogeny as well as their potential involvement in cortical area expansion and behavioral differences.

Strengths:

This article is very useful for comparative brain studies. It was conducted with great rigor and builds on numerous previous studies. The article is well written and very didactic. The different protocols for brain collection, perfusion and scanning are very detailed. The images are self-explanatory and of high quality. The authors explain their choice of nomenclature and labels for sulci and gyri on all species, with many arguments. The opening on ecology and social behavior in the discussion is of great interest and helps to put into perspective the differences in folding found at the level of the different cortexes. In addition, the authors do not forget to put their results into the context of the laws of allometry. They explain, for example, that although the largest brains were the most folded and had the deepest folds in their dataset, they did not necessarily have unique sulci, unlike some of the smaller, smoother brains.

Weaknesses:

Although an effort was made to take inter-individual variability into account, this approach could not be applied within each species, given the large number of wild animals. Sex differences could therefore not be analyzed either. However, this does not detract from the aim, which is to lay the foundations for a correspondence between the brains of carnivores in order to simplify navigation within the brains of these species for future studies. The authors also attempted to add measurements of sulcal length to this qualitative study, but it does not include other comparisons of morphometric data that are standard in sulci studies, such as sulcal depth, sulci wall surface area, or thickness of the cortical ribbon around the sulci.

---

## [Author Response]

The following is the authors’ response to the original reviews.

**Reviewer #1 (Public review):**
Summary:The paper by Boch and colleagues, entitled Comparative Neuroimaging of the Carnivore Brain: Neocortical Sulcal Anatomy, compares and describes the cortical sulci of eighteen carnivore species, and sets a benchmark for future work on comparative brains.Based on previous observations, electrophysiological, histological and neuroimaging studies and their own observations, the authors establish a correspondence between the cortical sulci and gyri of these species. The different folding patterns of all brain regions are detailed, put into perspective in relation to their phylogeny as well as their potential involvement in cortical area expansion and behavioral differences.Strengths:This is a pioneering article, very useful for comparative brain studies and conducted with great seriousness and based on many past studies. The article is well-written and very didactic. The different protocols for brain collection, perfusion, and scanning are very detailed. The images are self-explanatory and of high quality. The authors explain their choice of nomenclature and labels for sulci and gyri on all species, with many arguments. The opening on ecology and social behavior in the discussion is of great interest and helps to put into perspective the differences in folding found at the level of the different cortexes. In addition, the authors do not forget to put their results into the context of the laws of allometry. They explain, for example, that although the largest brains were the most folded and had the deepest folds in their dataset, they did not necessarily have unique sulci, unlike some of the smaller, smoother brains.Weaknesses:The article is aware of its limitations, not being able to take into account interindividual variability within each species, inter-hemispheric asymmetries, or differences between males and females. However, this does not detract from their aim, which is to lay the foundations for a correspondence between the brains of carnivores so that navigation within the brains of these species can be simplified for future studies. This article does not include comparisons of morphometric data such as sulci depth, sulci wall surface, or thickness of the cortical ribbon around the sulci.

We thank the reviewer for their overwhelmingly positive evaluation of our work. As noted by the reviewer, our primary aim was to establish a framework for navigating carnivoran brains to lay the foundation for future research. We are pleased that this objective has been successfully achieved.

Individual differences

As the reviewer points out, we do not quantify within-species intraindividual differences, which was a conscious choice. We aimed to emphasise the breadth of species over individuals, as is standard in large-scale comparative anatomy (cf. Heuer et al., 2023, eLife; Suarez et al., 2022, eLife). Following the logic of phylogenetic relationships, the presence of a particular sulcus across related species is also a measure of reliability. We felt safe in this choice, as previous work in both primates and carnivorans has shown that differences across major sulci across individuals are a matter of degree rather than a case of presence or absence (Connolly, 1950, External morphology of the primate brain, C.C. Thomas; Hecht et al., 2019 J Neurosci; Kawamuro 1971 Acta Anat., Kawamuro & Naito, 1977, Acta Anat.).

In our revised manuscript, we now include additional individuals for six different species, representing both carnivoran suborders (Feliformia and Caniformia), and within Caniformia, both Arctoidea and Canidae (see revised Table 1 and main changes in text below). These additions confirm that intra-species variation primarily affects sulcal shape rather than the presence or absence of major sulci. Furthermore, the inclusion of additional individuals helped validate some initial observations, for example, confirming that the brown bear's proreal sulcus is more accurately characterised as a branch of the presylvian sulcus.

Main changes in the revised manuscript:

Results and discussion, p. 13-14: Presylvian sulcus. Rostral to the pseudo-sylvian fissure, the perisylvian sulcus originates from or close to the rostral lateral rhinal fissure (see Supplementary Note 1 and Figure S2 for ventral view). The sulcus extends dorsally, and we observed a gentle caudal curve in the majority of the species (Figures 2-3, white).

There were no major variations across species, but we noted a shortened sulcus in the meerkat and Egyptian mongoose and the presence of a secondary branch at the dorsal end that extended rostrally in the Eurasian badger and South American coati brain. The brown bear exhibited an additional sulcus in the frontal lobe, previously labelled as the proreal sulcus (see, e.g., Sienkiewicz et al., 2019); however, its shape closely resembled the secondary branches of the perisylvian sulcus seen in the South American coati and Eurasian badger. Sienkiewicz et al. (2019) also noted that this sulcus merges with the presylvian sulcus in their specimen, consistent with our findings in the left hemisphere of the brown bear and bilaterally in the Ussuri brown bear (see Supplementary Figure S3A, S5A). Given the known gyrencephaly of Ursidae brains with frequent secondary and tertiary sulci (Lyras et al., 2023), we propose that this sulcus represents a branch of the perisylvian sulcus.

General Discussion, p. 23-24:Regarding individual variability in external brain morphology, previous work in primates and carnivorans has shown that differences across individuals typically affect sulcal shape, depth, or extent, but not the presence of major sulci. This has been reported in diverse contexts, including comparisons between captive and (semi-)wild macaque (Sallet et al., 2011; Testard et al., 2022), different dog breeds (Hecht et al., 2019), domestic cats (Kawamura, 1971b), or selectively bred foxes (Hecht et al., 2021). By including additional individuals for selected species, we extend these findings to a broader range of carnivorans. Notably, we observed no major sulcal differences between closely related species, even when specimens were acquired using different extraction and scanning protocols, for example, across felid clades or among wolf-like canids, further suggesting that substantial within-species variation is unlikely. While a full analysis of interindividual variability lies beyond the scope of this study, our findings support the reliability of the major sulcal patterns described.

Interhemispheric differences

Regarding potential inter-hemispheric differences, we have now also created digital atlases of all identified sulci in both hemispheres, which are publicly available at https://git.fmrib.ox.ac.uk/neuroecologylab/carnivore-surfaces. While the manuscript continues to focus primarily on descriptions of the right hemisphere, we now also report observed inter-hemispheric differences where applicable. These differences remain minor and, again, a matter of degree. For example, the complementary quantitative analyses investigating covariation between sulcal length and behavioural traits conducted in the right hemisphere were replicated in the left (Supplementary Figure S6 and related Supplementary tables S1-S3).

Main changes in the revised manuscript:

Materials and Methods, p. 33: We focused on the major lateral and dorsal sulci of the carnivoran brain, but the medial wall and ventral view of the sulci are also described. For consistency, we started by labelling the right hemispheres on the mid-thickness surfaces; these are the hemispheres presented in the manuscript. An exception was made for the jungle cat, for which only the left hemisphere was available and is therefore shown. We aimed to facilitate interspecies comparisons and the exploration of previously undescribed carnivoran brains. To this end, we first created standardized criteria (henceforth referred to as recipes) for identifying each sulcus, drawing from existing literature on carnivoran neuroanatomy, particularly in paleoneurology (Lyras et al., 2023), and our own observations. In addition, we created digital sulcal masks for both hemispheres, which allowed us to test whether the same patterns were observable bilaterally and to further facilitate future research building on our framework. For the Egyptian mongoose, only the right hemisphere was available, and thus, a bilateral comparison was not possible for this species. Anatomical nomenclature primarily follows the recommendations of Czeibert et al (2018); if applicable, alternative names of sulci are provided once.

Materials and Methods, p. 34-35: We first briefly illustrated the gyri of the carnivoran brain with a focus on gyri that are not present in some species as a consequence of absent sulci to complement our observations. We then summarised the key differences and similarities in sulcal anatomy between species and related them to their ecology and behaviour. To complement this qualitative description, we conducted an initial quantitative analysis of sulcal length data from both hemispheres.

To test whether sulcal length covaries with behavioural traits, we fit linear models predicting the relative length of the three target sulci (cruciate, postcruciate, proreal) as a function of forepaw dexterity (low vs. high) and sociality (solitary vs cooperative hunting). We measured the absolute length of each sulcus using the wb_command -border-length function from the Connectome Workbench toolkit (Marcus et al., 2011) applied to the manually defined sulcal masks (i.e., border files). Relative sulcal length was calculated by dividing the length of each target sulcus by that of a reference sulcus in the same hemisphere, reducing interspecies variation in brain or sulcal size. Reference sulci were required to be present in all species within a hemisphere and excluded if they were a target sulcus, part of the same functional system (e.g., somatosensory/motor), or anatomically atypical (e.g., the pseudosylvian fissure). This resulted in seven reference sulci for the proreal sulcus (ansate, coronal, marginal, presylvian, retrosplenial, splenial, suprasylvian) and four for the cruciate and postcruciate sulci (marginal, retrosplenial, splenial, suprasylvian). For each target-reference pair, we fit the following linear model: relative length ~ forepaw dexterity + sociality. Models were run separately for left and right hemispheres, with the left serving as a replication test. Associations were considered meaningful if the predictor reached statistical significance (p ≤ .05) in ≥ 75% of reference sulcus models per hemisphere. Additional individuals were not included in the analysis.

Data and code availability statement, p. 35-36: Generated surfaces of all species and T1-like contrast images of post-mortem samples obtained by the C Generated surfaces of all species and T1-like contrast images of post-mortem samples obtained by the Copenhagen Zoo and the Zoological Society of London (see Table 1) are available at the Digital Brain Zoo of the University of Oxford (Tendler et al., 2022) (https://open.win.ox.ac.uk/DigitalBrainBank/#/datasets/zoo). For all other species, except the domestic cat, the cortical surface reconstructions are available through the same resource. In-vivo data for the domestic cat is available upon request.

We created, extracted and analysed sulcal length data using the Connectome Workbench toolkit (Marcus et al., 2011), R 4.4.0 (R Core Team, 2023) and Python 3.9.7. Sulcal masks, along with the associated midthickness cortical surface reconstructions for all 32 animals, species-specific behavioural data, and the code used to extract sulcal lengths and perform the statistical analyses are available at: https://git.fmrib.ox.ac.uk/neuroecologylab/carnivore-surfaces.

Further brain measures

We feel that sulci depth, sulci wall surface, or thickness of the cortical ribbon are measures that vary more across individuals, and we have therefore not included them in the study. In addition, these are measures that are not generally used as betweenspecies comparative measures, whereas sulcal patterning is (cf. Amiez et al., 2019, Nat Comms; Connolly, 1950; Miller et al., 2021, Brain Behav Evol; Radinsky 1975, J Mammal; Radinsky 1969, Ann N Y Acad Sci; Welker & Campos 1963 J. Comp Neurol).

We, therefore, added them as suggestions for future directions, building on our work.

Major changes in the revised manuscript:

Limitations and future directions, p. 25-26: Our findings represent a critical first step for linking brains within and across species for interspecies insights. The present analyses are based on multiple individuals pooled into families and genera, primarily focusing on single representatives per species. Additional individuals for selected species confirmed that intra-species variation is a matter of degree rather than a case of presence or absence of major sulci, but we do not provide an extensive account of the possible range of sulcal shape or other anatomical features. Future studies will aim to systematically investigate interindividual variability in sulcal shape, depth, surface area, or thickness of the cortical ribbon surrounding the sulci, and will extend to more detailed investigations of the medial part of the cortex, as well as the subcortical structures and the cerebellum.The present framework and resulting database also provides the foundation to guide and facilitate future investigations of inter- and intra-species variation in regional brain size.

**Reviewer #2 (Public review):**
Summary:The authors have completed MRI-based descriptions of the sulcal anatomy of 18 carnivoran species that vary greatly in behaviour and ecology. In this descriptive study, different sulcal patterns are identified in relation to phylogeny and, to some extent, behaviour. The authors argue that the reported differences across families reflect behaviour and electrophysiology, but these correlations are not supported by any analyses.Strengths:A major strength of this paper is using very similar imaging methods across all specimens. Often papers like this rely on highly variable methods so that consistency reduces some of the variability that can arise due to methodology.The descriptive anatomy was accurate and precise. I could readily follow exactly where on the cortical surface the authors referring. This is not always the case for descriptive anatomy papers, so I appreciated the efforts the authors took to make the results understandable for a broader audience.I also greatly appreciate the authors making the images open access through their website.Weaknesses:Although I enjoyed many aspects of this manuscript, it is lacking in any quantitative analyses that would provide more insights into what these variations in sulcal anatomy might mean. The authors do discuss inter-clade differences in relation to behaviour and older electrophysiology papers by Welker, Campos, Johnson, and others, but it would be more biologically relevant to try to calculate surface areas or volumes of cortical fields defined by some of these sulci. For example, something like the endocast surface area measurements used by Sakai and colleagues would allow the authors to test for differences among clades, in relation to brain/body size, or behaviour. Quantitative measurements would also aid significantly in supporting some of the potential correlations hinted at in the Discussion.Although quantitative measurements would be helpful, there are also some significant concerns in relation to the specimens themselves. First, almost all of these are captive individuals. We know that environmental differences can alter neocortical development and humans and nonhuman animals and domestication affects neocortical volume and morphology. Whether captive breeding affects neocortical anatomy might not be known, but it can affect other brain regions and overall brain size and could affect sulcal patterns. Second, despite using similar imaging methods across specimens, fixation varied markedly across specimens. Fixation is unlikely to affect the ability to recognize deep sulci, but variations in shrinkage could nevertheless affect overall brain size and morphology, including the ability to recognize shallow sulci. Third, the sample size = 1 for every species examined. In humans and nonhuman animals, sulcal patterns can vary significantly among individuals. In domestic dogs, it can even vary greatly across breeds. It, therefore, remains unclear to what extent the pattern observed in one individual can be generalized for a species, let alone an entire genus or family. The lack of accounting for inter-individual variability makes it difficult to make any firm conclusions regarding the functional relevance of sulcal patterns.

We thank the reviewer for their assessment of our work. The primary aim of this study was to establish a framework for navigating carnivoran brains by providing a comprehensive overview of all major neocortical sulci across eighteen different species. Given the inconsistent nomenclature in the literature and the lack of standardized criteria (“recipes”) for identifying the major sulci, we specifically focused on homogenizing the terminology and creating recipes for their identification. In addition to generating digital cortical surfaces for all brains, we have now also added sulcal masks to further support future research building on this framework. We are pleased that our primary objective is seen as successfully achieved and are delighted to report that, following the reviewer’s recommendations, we have further expanded the dataset by including eight additional species and a second individual for six species, yielding a total of 32 carnivorans from eight carnivoran families (see revised Table 1 for a detailed list).

The present dataset constitutes the most comprehensive collection of fissiped carnivoran brains to date, encompassing a wide range of land-dwelling species from eight families. It includes diverse representatives, such as both social and solitary mongooses, weasel-like and non-weasel mustelids, and a broad spectrum of canids including wolf-like, fox-like, and more basal forms. Further expanding this already extensive dataset has even led to novel discoveries, such as the felid-specific diagonal sulcus and the unique occipito-temporal sulcal configuration shared by herpestids and hyaenids.

Major changes in the revised manuscript:

Results and discussion, p. 4-5: We labelled the neocortical sulci of twenty-six carnivoran species (see Figure 1) based on reconstructed surfaces and developed standardised criteria (“recipes”) for identifying each major sulcus. For each sulcus, we also created corresponding digital masks. Our study included eleven Feliformia and fifteen Caniformia species from eight different carnivoran families. Within the suborder Caniformia, we examined eight Canidae and seven Arctoidea species. In addition, we describe relative intra-species variation in sulcal shape based on supplementary specimens from six species (see Table 1).

Overall, of the carnivorans studied, Canidae brains exhibited the largest number of unique major sulci, while the brown bear brain was the most gyrencephalic, with the deepest folds and many secondary sulci (see Figures 2-3; brains are arranged by descending number of major sulci). The brown bear was also the largest animal in the sample. The brains of the smaller species, such as the fennec fox, meerkat or ferret, were the most lissencephalic, with the sulci having fewer undulations or indentations compared to the other species. A similar trend has also been observed in the sulci of the prefrontal cortex in primates (Amiez et al., 2023, 2019). The meerkat and Egyptian mongoose exhibited the smallest number of major sulci but possessed, along with the striped hyena, a unique configuration of sulci in the occipito-temporal cortex. In the following, we describe each sulcus' appearance, the recipes on how to identify them, and provide an overview of the most significant differences across species.

Results and discussion, p. 11: Diagonal sulcus. The diagonal sulcus is oriented nearly perpendicularly to the rostral portion of the suprasylvian sulcus (Figure 2, Supplementary Figure S2, red). We identified it in all Felidae and in the striped hyena, but it was absent in Herpestidae and all Caniformia species.

In our sample, the sulcus showed moderate variation in shape and continuity. In the caracal and the second sand cat, it appeared as a detached continuation of the rostral suprasylvian sulcus (Supplementary Figure S3). In the Amur and Persian leopards, the diagonal sulcus merged with the rostral ectosylvian sulcus on the right hemisphere, forming a continuous or bifurcated groove. Similar individual variation has been described in domestic cats (Kawamura, 1971b).

We respectfully disagree with the reviewer on two accounts, where we believe the revieweris not judging the scope of the current work

(1) Intra-individual differences & potential confounding factors

The first is with respect to individual differences relationships. To the best of our knowledge, differences between captive and wild animals, or indeed between individuals, do not affect the presence or absence of any major sulci. No differences in sulcal patterns were detected between captive and (semi-)wild macaques (cf. Sallet et al., 2011, Science; Testard et al., 2022, Sci Adv), different dog breeds (Hecht et al., 2019 J Neurosci) or foxes selectively bred to simulate domestication, compared to controls (Hecht et al., 2021 J. Neurosci).

By including additional individuals for selected species in the revised version of our manuscript, we confirm and extend these findings to a broader range of carnivorans. Indeed, we also did not observe major differences between closely related species, even when specimens were collected using different extraction and scanning protocols - for example, across felid clades or wolf-like canids - making substantial individual variation within a species even less likely. Thus, while a comprehensive analysis of interindividual variability is beyond the scope of this study, our observations support the robustness of the major sulcal patterns described here. Moreover, the inclusion of additional individuals also helped validate some initial observations, for example, confirming that the brown bear's proreal sulcus is more accurately characterised as a branch of the presylvian sulcus.

We do, however, agree with the reviewer that building up a database like ours benefits from providing as much information about the samples as possible to enable these issues to be tested. We, therefore, made sure to include as detailed information as possible, including whether the animals were from captive or wild populations, in our manuscript.

Main changes in the revised manuscript:

Results and discussion, p. 13-14: Presylvian sulcus. There were no major variations across species, but we noted a shortened sulcus in the meerkat and Egyptian mongoose and the presence of a secondary branch at the dorsal end that extended rostrally in the Eurasian badger and South American coati brain. The brown bear exhibited an additional sulcus in the frontal lobe, previously labelled as the proreal sulcus (see, e.g., Sienkiewicz et al., 2019); however, its shape closely resembled the secondary branches of the perisylvian sulcus seen in the South American coati and Eurasian badger. Sienkiewicz et al. (2019) also noted that this sulcus merges with the presylvian sulcus in their specimen, consistent with our findings in the left hemisphere of the brown bear and bilaterally in the Ussuri brown bear (see Supplementary Figure S3A, S5A). Given the known gyrencephaly of Ursidae brains with frequent secondary and tertiary sulci (Lyras et al., 2023), we propose that this sulcus represents a branch of the perisylvian sulcus.

Results and discussion, p. 23-24: Regarding individual variability in external brain morphology, previous work in primates and carnivorans has shown that differences across individuals typically affect sulcal shape, depth, or extent, but not the presence of major sulci. This has been reported in diverse contexts, including comparisons between captive and (semi-)wild macaque (Sallet et al., 2011; Testard et al., 2022), different dog breeds (Hecht et al., 2019), domestic cats (Kawamura, 1971b), or selectively bred foxes (Hecht et al., 2021). By including additional individuals for selected species, we extend these findings to a broader range of carnivorans. Notably, we observed no major sulcal differences between closely related species, even when specimens were acquired using different extraction and scanning protocols, for example, across felid clades or among wolf-like canids, further suggesting that substantial within-species variation is unlikely. While a full analysis of interindividual variability lies beyond the scope of this study, our findings support the reliability of the major sulcal patterns described.

Limitations and future directions, p. 25-26: Our findings represent a critical first step for linking brains within and across species for interspecies insights. The present analyses are based on multiple individuals pooled into families and genera, primarily focusing on single representatives per species. Additional individuals for selected species confirmed that intra-species variation is a matter of degree rather than a case of presence or absence of major sulci, but we do not provide an extensive account of the possible range of sulcal shape or other anatomical features.

Future studies will aim to systematically investigate interindividual variability in sulcal shape, depth, surface area, or thickness of the cortical ribbon surrounding the sulci, and will extend to more detailed investigations of the medial part of the cortex, as well as the subcortical structures and the cerebellum.The present framework and resulting database also provides the foundation to guide and facilitate future investigations of inter- and intra-species variation in regional brain size.

(2) Quantification of structure/function relationships

The second is in the quantification of structure/function relationships. We believe the cortical surfaces, detailed sulci descriptions, and atlases themselves are the main deliverables of this project. We felt it prudent to include some qualitative descriptions of the relationship between sulci as we observed them and behaviours as known from the literature, as a way to illustrate the possibilities that this foundational work opens up. This approach also allowed us to confirm and extend previous findings based on observations from a less diverse range of carnivoran species and families (Radinsky 1968 J Comp Neurol; Radinsky 1969, Ann N Y Acad Sci; Welker & Campos 1963 J Comp Neurol; Welker & Seidenstein, 1959 J Comp Neurol).

However, a full statistical framework for analysis is beyond the scope of this paper. Our group has previously worked on methods to quantitatively compare brain organization across species - indeed, we have developed a full framework for doing so (Mars et al., 2021, Annu Rev Neurosci), based on the idea that brains that differ in size and morphology should be compared based on anatomical features in a common feature space. Previously, we have used white matter anatomy (Mars et al., 2018, eLife) and spatial transcriptomics (Beauchamp et al., 2021, eLife). The present work presents the foundation for this approach to be expanded to sulcal anatomy, but the full development of it will be the topic of future communications.

Nevertheless, we now include a preliminary quantitative analysis of the relationship between the relative length of specific sulci and the two behavioural traits of interest. These analyses, which complement the qualitative observations in Figure 5, show that the relative length of the proreal sulcus was consistently greater in highly social, cooperatively hunting species, while no effect of forepaw dexterity was found (Supplementary Table S1). In contrast, both the cruciate and postcruciate sulci were significantly longer in species with high forepaw dexterity, but not related to sociality (Supplementary Tables S2–S3). These findings were consistent across reference sulci used to compute relative sulcal length and replicated in the left hemisphere (see Supplementary Figure S6).

We also would like to emphasize that we strongly believe that looking at measures of brain organization at a more detailed level than brain size or relative brain size is informative. Although studies correlating brain size with behavioural variables are prominent in the literature, they often struggle to distinguish between competing behavioural hypotheses (Healy, 2021, Adaptation and the Brain, OUP). In contrast, connectivity has a much more direct relationship to behavioural differences across species (Bryant et al., 2024, JoN), as does sulcal anatomy (Amiez et al., 2019, Nat Comms; Miller et al., 2021, Brain Behav Evol). Using our sulcal framework, we observed lineage-specific variations that would be overlooked by analyses focused solely on brain size. Moreover, such measures are less sensitive to the effects of fixation since that will affect brain size but not the presence or absence of a sulcus.

Main changes in the revised manuscript:

Results and discussion, p. 16-17: In the raccoon, red panda, coati, and ferret, considerably larger portions of the postcruciate gyrus S1 area appeared to be allocated to representing the forepaw and forelimbs (McLaughlin et al., 1998; Welker and Campos, 1963; Welker and Seidenstein, 1959) when compared to the domestic cat or dog (Dykes et al., 1980; Pinto Hamuy et al., 1956). This aligns with the observation that all species in the present sample with more complex or elongated postcruciate and cruciate sulci configurations display a preference for using their forepaws when manipulating their environment (see e.g., Iwaniuk et al., 1999; Iwaniuk and Whishaw, 1999; Radinsky, 1968; and Figure 5A). Complementary quantitative analyses further support this link, revealing a positive relationship between the relative length of the cruciate and postcruciate sulci and high forepaw dexterity (see Supplementary Figure S6, Tables S2-S3). This is suggestive of a potential link between sulcal morphology and a behavioural specialization in Arctoidea, consistent with earlier observations in otter species (Radinsky, 1968).

Results and discussion, p. 21: A distinct proreal sulcus was observed in the frontal lobe of the domestic dog, the African wild dog, wolf, dingo, and bush dog. This may indicate an expansion of frontal cortex in these animals compared to the other species in our sample (Figure 5-6). This aligns with findings from a comprehensive study comparing canid endocasts revealing an expanded proreal gyrus in these animals compared to the fennec fox, red fox and other species of the genus Vulpes (Lyras and Van Der Geer, 2003). The canids with a proreal sulcus also exhibit complex social structures compared to the primarily solitary living foxes (Nowak, 2005; Wilson and Mittermeier, 2009; Wilson, 2000, and see Figure 5).Despite living in social groups, the bat-eared fox, an insectivorous canid, does not possess a proreal sulcus. Its foraging behaviour is best described as spatially or communally coordinated rather than truly cooperative (Macdonald and Sillero-Zubiri, 2004), suggesting that the relationship between sulcal morphology and sociality may be specific to species engaging in active cooperative hunting. Supplementary quantitative analyses also confirm an increase in the relative length of the proreal sulcus

in cooperatively hunting species Moreover, a previous investigation of Canidae and Felidae brain evolution, using endocasts of extant and extinct species, also suggested a link between the emergence of pack structures and the proreal sulcus in Canidae (Radinsky, 1969). Despite being highly social and living in large social groups (i.e., mobs), meerkats appear to have a relatively small frontal lobe and no proreal sulcus compared to the social Canids (Figure 5), which would suggest that if the presence of a proreal sulcus correlates with complex social behaviour, this is canid-specific.

General discussion, p. 22-23: Our results revealed several interesting patterns of local variation in sulcal morphology between and within different lineages, and successfully replicate and expand upon prior observations based on more limited sets of species (Radinsky, 1969, 1968; Welker and Campos, 1963; Welker and Seidenstein, 1959). For example, Arctoidea showed relatively complex sulcal anatomy in the somatosensory cortex but low complexity in the occipito-temporal regions. In Canidae and Felidae, we found more complex occipito-temporal sulcal patterns indicative of changes in the amount of cortex devoted to visual and auditory processing in these regions. These observations may be linked to social or ecological factors, such as how the animals interact with objects or each other and their varied foraging strategies. Another example was the differential relative expansion of the neocortex surrounding the cruciate sulcus, which was particularly complex in Arctoidea species that are known to use their paws to manipulate their environment. Consistent with this observation, complementary quantitative analyses of both hemispheres revealed that species with high forepaw dexterity tended to have longer cruciate and postcruciate sulci. Although it has been argued that the cruciate sulcus appeared independently in different lineages and its exact relationship to the location of primary motor areas varies (Radinsky, 1971), our results provide a detailed exploration of the relationship between brain morphology and behavioural preferences across such a range of species.

Materials and Methods, p. 33: We focused on the major lateral and dorsal sulci of the carnivoran brain, but the medial wall and ventral view of the sulci are also described. For consistency, we started by labelling the right hemispheres on the mid-thickness surfaces; these are the hemispheres presented in the manuscript. An exception was made for the jungle cat, for which only the left hemisphere was available and is therefore shown. We aimed to facilitate interspecies comparisons and the exploration of previously undescribed carnivoran brains. To this end, we first created standardized criteria (henceforth referred to as recipes) for identifying each sulcus, drawing from existing literature on carnivoran neuroanatomy, particularly in paleoneurology (Lyras et al., 2023), and our own observations.In addition, we created digital sulcal masks for both hemispheres, which allowed us to test whether the same patterns were observable bilaterally and to further facilitate future research building on our framework. For the Egyptian mongoose, only the right hemisphere was available, and thus, a bilateral comparison was not possible for this species. Anatomical nomenclature primarily follows the recommendations of Czeibert et al (2018); if applicable, alternative names of sulci are provided once.

Materials and Methods, p. 34-35: We first briefly illustrated the gyri of the carnivoran brain with a focus on gyri that are not present in some species as a consequence of absent sulci to complement our observations. We then summarised the key differences and similarities in sulcal anatomy between species and related them to their ecology and behaviour. To complement this qualitative description, we conducted an initial quantitative analysis of sulcal length data from both hemispheres. To test whether sulcal length covaries with behavioural traits, we fit linear models predicting the relative length of the three target sulci (cruciate, postcruciate, proreal) as a function of forepaw dexterity (low vs.high) and sociality (solitary vs cooperative hunting). We measured the absolute length of each sulcus using the wb_command -border-length function from the Connectome Workbench toolkit (Marcus et al., 2011) applied to the manually defined sulcal masks (i.e., border files). Relative sulcal length was calculated by dividing the length of each target sulcus by that of a reference sulcus in the same hemisphere, reducing interspecies variation in brain or sulcal size. Reference sulci were required to be present in all species within a hemisphere and excluded if they were a target sulcus, part of the same functional system (e.g., somatosensory/motor), or anatomically atypical (e.g., the pseudosylvian fissure). This resulted in seven reference sulci for the proreal sulcus (ansate, coronal, marginal, presylvian, retrosplenial, splenial, suprasylvian) and four for the cruciate and postcruciate sulci (marginal, retrosplenial, splenial, suprasylvian). For each target-reference pair, we fit the following linear model: relative length ~ forepaw dexterity + sociality. Models were run separately for left and right hemispheres, with the left serving as a replication test. Associations were considered meaningful if the predictor reached statistical significance (p ≤ .05) in ≥ 75% of reference sulcus models per hemisphere. Additional individuals were not included in the analysis.

Data and code availability statement, p. 35-36: Generated surfaces of all species and T1-like contrast images of post-mortem samples obtained by the C Generated surfaces of all species and T1-like contrast images of post-mortem samples obtained by the Copenhagen Zoo and the Zoological Society of London (see Table 1) are available at the Digital Brain Zoo of the University of Oxford (Tendler et al., 2022) (https://open.win.ox.ac.uk/DigitalBrainBank/#/datasets/zoo). For all other species, except the domestic cat, the cortical surface reconstructions are available through the same resource. In-vivo data for the domestic cat is available upon request.

We created, extracted and analysed sulcal length data using the Connectome Workbench toolkit (Marcus et al., 2011), R 4.4.0 (R Core Team, 2023) and Python 3.9.7. Sulcal masks, along with the associated midthickness cortical surface reconstructions for all 32 animals, species-specific behavioural data, and the code used to extract sulcal lengths and perform the statistical analyses are available at:

https://git.fmrib.ox.ac.uk/neuroecologylab/carnivore-surfaces.

**Reviewer #1 (Recommendations for the authors):**
I was convinced by your model of labels in the temporal region and the nomenclature used, thanks to your argument concerning the primary auditory area in ferrets located in the gyrus called ectosylvian even though they have no ectosylvian sulcus. While this region raises questions, it seems to me that you make a good case for your labelling.However, I don't understand your arguments in the occipital region regarding the ectomarginal sulcus. In the bear, for example, I don't understand why the caudal part of the marginal sulcus is not referred to as ectomarginal? You say that this sulci is specific to canids.Whether in the paragraph describing the ectomarginal sulcus, the marginal sulcus, in the paragraphs on the gyri, or in the paragraph concerning the potential relationship to function, I don't see any argument to support your hypothesis. Especially as there is no information in the literature on the functions in this area of the bear brain as in that of the dog or other related species.You just mention that in Canidae, the ectomarginal "runs between the suprasylvian and marginal sulcus", and I don't see why this is an argument.Could you explain in more detail your choice of label and the specificity you claim to have in the canids of this region?

We have now expanded our rationale in the revised manuscript, particularly in the section describing the marginal sulcus, which directly follows the description of the ectomarginal sulcus. In brief, across our sample, including Ursidae and Canidae, we observed variation in whether the caudal marginal sulcus was detached or continuous, or extended further caudally vs ventrally, but no separate additional sulcus resembling the ectomarginal sulcus was seen in any species outside the canid family. We therefore reserve the label ectomarginal sulcus for the distinct structure consistently observed in Canidae and avoid applying it to the detached caudal marginal sulcus observed in Ursidae.

Main changes in the revised manuscript:

Results and discussion, p. 10-11: In several species, including the dingo, domestic cat, brown bear and South American coati and further supplementary individuals (Supplementary figure S3B), the caudal portion of the marginal sulcus was detached in one or both hemispheres, which is a frequently reported occurrence (England, 1973; Kawamura, 1971a; Kawamura and Naito, 1978). Potentially due to the similar caudal bend, some authors have labelled the (detached) caudal portion of the marginal sulcus in Ursidae as the ectomarginal sulcus (Lyras et al., 2023, but see e.g., Sienkiewicz et al., 2019);

The (detached) caudal marginal sulcus in Ursidae continues the course of the marginal sulcus caudally and/or ventrally and is topologically continuous with it. In contrast, the ectomarginal sulcus in Canidae is an entirely separate sulcus that runs between the suprasylvian and marginal sulci, forming a small, additional arch that is rarely connected to the marginal sulcus (Kawamura and Naito, 1978). This distinction is illustrated, for example, in the dingo and grey wolf. In the dingo, we observed both a detached caudal extension of the marginal sulcus and a distinct ectomarginal sulcus. In both grey wolf specimens, the marginal sulcus extended ventrally in a way that resembled the brown bear, but they also exhibited a clearly separate ectomarginal sulcus, confirming that the two features are not equivalent. In contrast, in the brown bear and Ussuri brown bear (Supplementary Figure S3B), we observed variation in whether the marginal sulcus was detached or continuous, but no separate sulcus resembling the ectomarginal sulcus seen in Canidae.

**Reviewer #2 (Recommendations for the authors):**
Although I indicated this already, I stress that the lack of quantification is problematic. In its current format, this is a classic descriptive study suitable for an anatomy journal, but even then, the conclusions are highly speculative. I would advise including some quantification of sulcal lengths or depths and surface areas or volumes of individual regions and relate all of those to overall brain size and potential clade differences. Figure 5 hints at some of these putative correlations, but is not an analysis. Some of these correlations are discussed in the manuscript, but without quantification, it is simply more descriptions and some speculative associations that largely parallel and corroborate findings from Radinsky's papers. In addition to quantification, the authors should consider a more fulsome explanation of the potential confounds and limitations of their data. As alluded to above, there are many sources of variation that were not sufficiently discussed but are critically important for interpreting any putative differences among and within clades.

We would like to reiterate that the primary aim of our study was to establish a comprehensive sulcal framework for carnivoran brains. The behavioural and ecological associations were secondary and exploratory, arising from a first application of this framework, and will require further investigation in future studies.

We already acknowledged in the initial version of the manuscript that many of our observations were consistent with those previously reported by Radinsky in more limited sets of species. However, we recognise that this point may not have come across clearly. We carefully revised our manuscript to further emphasise that our findings replicate and extend Radinsky’s work in a larger cross-species comparison, showing that our framework also successfully replicates and expands prior work.

As detailed in the public reviews, we did not measure overall or relative brain sizes. However, in the revised version of the manuscript, we have now quantified the relationship between sulcal length and its association with forepaw dexterity and sociality to complement the qualitative observations in Figure 5. Although preliminary, we believe that these analyses further showcase the strength of our sulcal framework and its potential for future investigations.

We also revised our discussion section to highlight the potential for future studies to build on our framework to systematically investigate interindividual variability in sulcal shape, depth, surface area, or thickness of the cortical ribbon surrounding the sulci. We also added that our framework and accompanying dataset can facilitate and guide future investigations into both inter- and intra-species variation in regional brain size.

Main changes in the revised manuscript:

General discussion, p. 22-23: Our results revealed several interesting patterns of local variation in sulcal morphology between and within different lineages, and successfully replicate and expand upon prior observations based on more limited sets of species (Radinsky, 1969, 1968; Welker and Campos, 1963; Welker and Seidenstein, 1959). For example, Arctoidea showed relatively complex sulcal anatomy in the somatosensory cortex but low complexity in the occipito-temporal regions. In Canidae and Felidae, we found more complex occipito-temporal sulcal patterns indicative of changes in the amount of cortex devoted to visual and auditory processing in these regions. These observations may be linked to social or ecological factors, such as how the animals interact with objects or each other and their varied foraging strategies. Another example was the differential relative expansion of the neocortex surrounding the cruciate sulcus, which was particularly complex in Arctoidea species that are known to use their paws to manipulate their environment. Consistent with this observation, complementary quantitative analyses of both hemispheres revealed that species with high forepaw dexterity tended to have longer cruciate and postcruciate sulci. Although it has been argued that the cruciate sulcus appeared independently in different lineages and its exact relationship to the location of primary motor areas varies (Radinsky, 1971), our results provide a detailed exploration of the relationship between brain morphology and behavioural preferences across such a range of species.

Limitations and future directions, p. 25-26: Our findings represent a critical first step for linking brains within and across species for interspecies insights. The present analyses are based on multiple individuals pooled into families and genera, primarily focusing on single representatives per species. Additional individuals for selected species confirmed that intra-species variation is a matter of degree rather than a case of presence or absence of major sulci, but we do not provide an extensive account of the possible range of sulcal shape or other anatomical features. Future studies will aim to systematically investigate interindividual variability in sulcal shape, depth, surface area, or thickness of the cortical ribbon surrounding the sulci, and will extend to more detailed investigations of the medial part of the cortex, as well as the subcortical structures and the cerebellum. The present framework and resulting database also provides the foundation to guide and facilitate future investigations of inter- and intra-species variation in regional brain size.

Another point that I did not see raised in the Discussion, but would be important and useful to include is that the authors are lacking specimens for several clades that could show additional differences in neocortical anatomy. For example, no hyaenids or viverrids were represented and an otter and badger are not necessarily representative of all mustelids, the majority of which are weasel-like. One could even argue that the meerkat is not necessarily representative of all herpestids given its behaviour and ecology. Of course, there are also pinnipeds, but they are divergent in many ways, and restricting the analyses to fissiped carnivorans is completely reasonable. Please note that I am not suggesting that the authors go back and try to procure even more species; rather they should emphasize that this is an incomplete survey of fissiped carnivorans.

The reviewer’s comments prompted us to further expand our carnivoran brain collection to include a broader range of species, representatives, and individual specimens. Notably, the collection now includes a hyaenid representative, the striped hyena. In addition to the otter and badger, we have added a weasel-like mustelid, the ferret, as well as the solitary Egyptian mongoose to complement the highly social meerkat within Herpestidae. Our felid dataset has also been expanded to include additional small and large wild cats, such as the sand cat and the Bengal tiger. As described above, these additions have led to the discovery of novel sulcal patterns, including the felid-specific diagonal sulcus.

We now also specify the fissiped families currently missing from the collection, which can be readily incorporated using our existing sulcal framework. The same applies to pinniped species, which we are currently investigating to support broader macro-level comparisons across the order.

Main changes in the revised manuscript:

General discussion, p. 23: Comparative neuroimaging requires balancing the level of anatomical detail with the breadth of species. The present sample represents the most comprehensive collection of fissiped carnivoran brains to date, encompassing a wide range of land-dwelling species from eight families. It includes diverse representatives, such as both social and solitary mongooses, weasel-like and non-weasel mustelids, and a broad array of canids, including wolf-like, fox-like, and more basal forms of canids. The framework and detailed protocols developed in this study are designed to facilitate navigation of additional fissiped species, such as Viverridae, Eupleridae, Mephitidae, Nandiniidae, and

Prionodontidae. Moreover, the approach can be readily extended to aquatic carnivorans, enabling broader macro-level comparisons across the order.

Apart from these broader issues, I also found some of the figures difficult to interpret in many instances. For example, the colour scheme used to highlight sulci is not colourblind friendly for Figures 2 and 3. It was also difficult for me to glean much information from Figure 6. I understand that functional regions of the cortex are shown for those species that were subject to electrophysiological studies in the past, but I could not work out how to transfer that data to the other brains. One suggestion for improving this would be to highlight putative cortical regions on the other brains in a lighter shade of the same colours.

We have carefully revised our figures to improve clarity and accessibility, particularly for individuals with colour vision deficiencies. Specifically, we have added numerical labels alongside the coloured sulci labels in Figures 2 and 3, as well as in all related supplementary figures (see examples on the following pages). For sulci that merge, such as the marginal, ansate, and coronal sulci, we have used colour combinations that are distinguishable across all major types of colour-blindness. Figure 4 has also been updated with a colour-blind-friendly palette and additional numerical labels for the gyri to further enhance interpretability.

Regarding Figure 6, we have updated the colour palette to ensure accessibility and have labelled all landmark sulci discussed in the main text using acronyms (e.g., the postcruciate sulcus as the boundary between S1 and M1). This is intended to facilitate the transfer of information between brains and guide orientation for readers less familiar with these structures. While we appreciate the suggestion to highlight putative cortical regions on other brains, we have opted not to do so. Our concern is that such visual cues, even when rendered in lighter shades, may be misinterpreted as established rather than hypothetical regional boundaries. We believe this more conservative approach appropriately reflects the current evidence base and avoids unintentionally overstating the certainty of functional homologies.